# Feto-maternal cholesterol transport regulated by β-Klotho–FGF15 axis is essential for fetal growth

Kanako Kobayashi[1,2], Kazuko Iwasa[1,2], Rika Azuma-Suzuki[2], Takeshi Kawauchi[1,2,3,4], Yo-ichi Nabeshima[1,2]

**β-Klotho (β-KL) is indispensable to regulate lipid, glucose, and energy metabolism in adult animals. β-KL is highly expressed in the yolk sac, but its role in the developmental stages has not been established. We hypothesized that β-KL is required for metabolic regulation in the embryo and aimed to clarify the role of β-KL during development. Here, we show that β-KL regulates feto-maternal cholesterol transport through the yolk sac by mediating FGF 15 signaling, and also that impairment of the β-KL–FGF15 axis causes fetal growth restriction (FGR). Embryos of β-kl knockout (β-kl−/−) mice were morphologically normal but exhibited FGR before placental maturation. The body weight of β-kl−/− mice remained lower after birth. β-KL deletion reduced cholesterol supply from the maternal blood and led to lipid shortage in the embryos. These phenotypes were similar to those of embryos lacking FGF15, indicating that β-KL–FGF15 axis is essential for growth and lipid regulation in the embryonic stages. Our findings suggest that lipid abnormalities in early gestation provoke FGR, leading to reduced body size in later life.**

## Introduction

Fetal growth restriction (FGR) is a pregnancy complication associated with a higher risk of perinatal morbidity, mortality, and poor health outcomes in infants (Unterscheider & Cuzzilla, 2021). Inadequate nutrition in utero is one of the major causes of FGR. Before the placenta is sufficiently established, embryos in early gestation absorb maternal nutrients via histotrophic mechanisms in the yolk sac (Bielinska et al, 1999; Zohn & Sarkar, 2010). For example, tracer experiments using stable isotope-labeled cholesterol showed that cholesterol derived from maternal blood is transported to embryos through the yolk sac in mice (Yoshida & Wada, 2005). The yolk sac is a membranous sac that derives from the hypoblast, an extra-embryonic lineage originating from the early inner cell mass of the blastocyst. Importantly, the yolk sac-dependent period overlaps with organogenesis in humans and mice.

Although mechanisms of nutrient transport through the placenta have been intensively investigated (Winterhager & Gellhaus, 2017; Chassen & Jansson, 2020), less attention has been paid to the yolk sac than to the placenta in the context of FGR.

β-Klotho (β-KL) is a type-I transmembrane protein expressed in the liver, exocrine pancreas, adipose tissues, and brain (Ito et al, 2005; Tomiyama et al, 2010; Ding et al, 2012; Bookout et al, 2013; Owen et al, 2013; Coate et al, 2017). Consistent with its high abundance in metabolic tissues, β-KL regulates lipid, glucose, and energy homeostasis by mediating FGF15 or FGF21 signaling in a tissue-specific manner (Owen et al, 2015). In the liver, β-KL serves as a coreceptor of FGF receptor 4 (FGFR4) to suppress cholesterol and bile acid biosynthesis via FGF15 signaling from the ileum (Inagaki et al, 2005; Ito et al, 2005; Yu et al, 2005; Tomiyama et al, 2010). Consistent with β-KL's critical roles in metabolism, β-kl knockout (β-kl−/−) mice exhibit reduced body weight and a shorter body length (Ito et al, 2005; Kobayashi et al, 2016; Somm et al, 2017), though a line of studies using whole-body or tissue-specific β-kl−/− mice showed that β-KL expressed in the liver, adipose tissues, and brain does not contribute to body weight regulation (Ding et al, 2012; Bookout et al, 2013; Kobayashi et al, 2016). Interestingly, growth restriction was observed in β-kl−/− embryos at E14.5 (Somm et al, 2018), suggesting that β-KL is required for fetal growth. During embryonic development, β-kl is strongly expressed in the visceral endoderm of the yolk sac in mice (Ito et al, 2000); however, the developmental role of β-KL has not been established.

We hypothesized that β-KL expressed in the yolk sac is essential for metabolic regulation in the embryo, and consequently, would have an impact on fetal growth and body size in postnatal life. To test this hypothesis, we analyzed β-kl−/− embryos at the transcript and metabolite levels. We found that the size of β-kl−/− embryos became significantly smaller than that of their control littermates during the yolk sac-dependent period. RNA-seq analysis and tracer experiments revealed that β-KL is required for the transport of cholesterol from the mother to the embryos through the yolk sac. Moreover, we showed that the regulation of growth and lipid metabolism by yolk sac β-KL is mediated by FGF15 signaling from the embryo.

[1]Department of Aging Science and Medicine, Kyoto University Graduate School of Medicine, Kyoto, Japan   [2]Laboratory of Molecular Life Science, Institute of Biomedical Research and Innovation, Foundation for Biomedical Research and Innovation at Kobe, Kobe, Japan   [3]Department of Adaptive and Maladaptive Responses in Health and Disease, Kyoto University Graduate School of Medicine, Kyoto, Japan   [4]Department of Physiology, Keio University School of Medicine, Tokyo, Japan

Correspondence: kobayashi.kanako.8z@kyoto-u.ac.jp; nabeshima.yoichi.7n@kyoto-u.ac.jp

# Results

## β-KL is required for growth in postimplantation stages

To evaluate the impact of β-KL on growth, we examined the size of β-kl−/− mice through the developmental stages to the postnatal period. We first compared the size and cell numbers of blastocysts at E3.5 among the genotypes. Before implantation, no difference was observed in the size or number of cells in β-kl−/− blastocysts compared with β-kl+/+ and β-kl+/− blastocysts (Fig 1A and B).

Murine embryos absorb maternal nutrients through the yolk sac until E10 (Zohn & Sarkar, 2010). Hence, we next examined the size of embryos at E9.5 as a representative point in the yolk sac-dependent period. The crown-rump length (CRL) of β-kl−/− embryos was significantly shorter than that of β-kl+/− embryos at E9.5 (Fig 1C). The somite numbers were comparable among the three genotypes (Fig 1D). Because β-kl+/+ and β-kl+/− embryos were indistinguishable in terms of growth at E3.5 and E9.5, we used β-kl+/− embryos as a control to provide a sufficient number of embryos for experiments. We again confirmed that β-kl−/− embryos were significantly smaller than β-kl+/− embryos in different cohorts (Fig S1A). The body weight of β-kl−/− mice was lower in both female and male animals after birth (Ito et al, 2005). Likewise, no sex bias was observed in CRL at E9.5 (Fig S1B and C). On the other hand, the birth ratio of β-kl−/− mice was less than expected from Mendelian inheritance (Ito et al, 2005; Somm et al, 2017), though the embryonic genotypes were segregated to approximately Mendelian ratios at E9.5 (Table S1). The gross appearance of β-kl−/− embryos was normal at E9.5 (Fig S1D). We also evaluated the CRL at E12.5 and confirmed that β-kl−/− embryos remained smaller than β-kl+/− embryos during the placenta-dependent period (Fig 1E).

Consistent with our previous report (Kobayashi et al, 2016), the body weight of β-kl−/− mice remained significantly lower than that of their control littermates (Fig 1F). These findings demonstrate that β-KL is required for growth in the postimplantation embryos, but is not involved in morphogenesis or survival up to E9.5.

## β-KL deletion leads to lipid abnormalities in the E9.5 embryo and yolk sac

To gain insight into the mechanisms underlying growth restriction in β-kl−/− embryos, we performed transcriptome analysis by RNA-sequencing. Among 16,314 genes significantly changed in β-kl−/− embryos compared with β-kl+/− embryos, the top 2,000 variable genes were divided into four groups using k-means clustering. The genes in cluster C, which is the cluster of genes related to cholesterol and sterol metabolism, were separated by genotypes (Fig 2A and Table S2). Another pathway analysis by the GAGE method also showed that sterol biosynthesis and metabolic processes were activated in β-kl−/− embryos (Fig 2B and Table S3). Quantitative real-time PCR (qRT-PCR) analyses confirmed that genes involved in lipid metabolism were up-regulated in β-kl−/− compared with β-kl+/− embryos (Fig 2C).

To judge the significance of changes in transcript levels, we compared the expression levels of genes related to lipid metabolism in fetal tissues from control β-kl+/− embryos with those in adult livers (Fig S2A). At E9.5, the level of 3-hydroxy-3-methylglutaryl-CoA reductase (Hmgcr) was higher in embryos than in adult liver, whereas 3-hydroxy-3-methylglutaryl-CoA synthase 1 (Hmgcs1) and the fatty acid synthase (Fasn) levels were comparable (Fig S2B). Low-density lipoprotein (LDL) receptor (Ldlr) expression in embryos was comparable with that in adult liver (Fig S2C). In contrast to Ldlr, high-density lipoprotein receptor (Scarb1) expression was markedly higher in yolk sacs than in embryos (Fig S2C).

We also examined the mRNA levels of enzymes in synthesizing bile acids because β-KL is indispensable for suppressing bile acid synthesis in adult mice (Ito et al, 2005; Tomiyama et al, 2010; Kobayashi et al, 2016). In the liver of adult β-kl−/− mice, the expression of Cyp7a1, a rate-limiting enzyme of hepatic bile acid synthesis, is strongly up-regulated, but Cyp7a1 expression was undetectable in E9.5 embryos and yolk sacs by qRT-PCR (data not shown). It is believed that the alternative pathway is the main contributor to production of bile acids in the fetal liver (Jia et al, 2021). However, to our surprise, the level of Cyp7b1, a gatekeeper to the alternative pathway, was much lower in E9.5 embryos than in the adult liver (Fig S2D). Furthermore, Cyp7b1 was expressed at similar levels in β-kl−/− and β-kl+/− embryos (Fig S2E), contrary to our expectation that β-KL would be required to suppress the alternative pathway in the embryo.

Transcription of Hmgcr and Ldlr is induced when cellular cholesterol levels are low (Silva Afonso et al, 2018). Therefore, we next evaluated lipid levels in E9.5 embryos and yolk sacs by TLC. The levels of cholesterol esters (CE) and triglycerides (TG) per μg of protein were significantly decreased in β-kl−/− embryos compared with their control littermates (Fig 2D). In contrast to the embryos, CE and TG were both increased in β-kl−/− yolk sacs (Fig 2E). The murine yolk sac is a membranous sac surrounding the embryo (Zohn & Sarkar, 2010), and its size as a tissue size is smaller than that of the whole embryo. Indeed, we confirmed this in the TLC samples by using total protein amount as a surrogate indicator (Fig S3A). Surprisingly, however, the CE content per tissue was higher in the yolk sacs than in the embryos, whereas TG levels were comparable in yolk sacs and embryos (Fig S3B). β-kl−/− yolk sacs were significantly smaller than β-kl+/− yolk sacs (Fig S3A), but no difference was observed in the lipid levels per yolk sacs (Fig S3B). Whether assessed per μg of protein or per tissue, β-kl−/− embryos showed significantly reduced lipid levels (Figs 2D and S3B). We also measured the lipid contents in the placenta, because β-kl mRNA was detected in the E12.5 placenta, even though at markedly lower levels than in the yolk sac or fetal liver (Fig S3C). We found that the lipid contents were similar in β-kl−/− and β-kl+/− placentas (Fig S3D). These results indicate that β-KL is required to regulate lipid metabolism in the embryo in the yolk sac-dependent period.

## β-KL is required to regulate maternal cholesterol transport through the yolk sac

To better understand the role of β-KL in lipid metabolism during development, we focused on β-KL expression in the yolk sac. At E9.5, the level of β-kl mRNA in the yolk sac was ~130-fold higher than that in the whole embryo (Fig 3A). Notably, β-kl expression in the yolk sac was also 1.8-fold higher than that in adult liver. We also

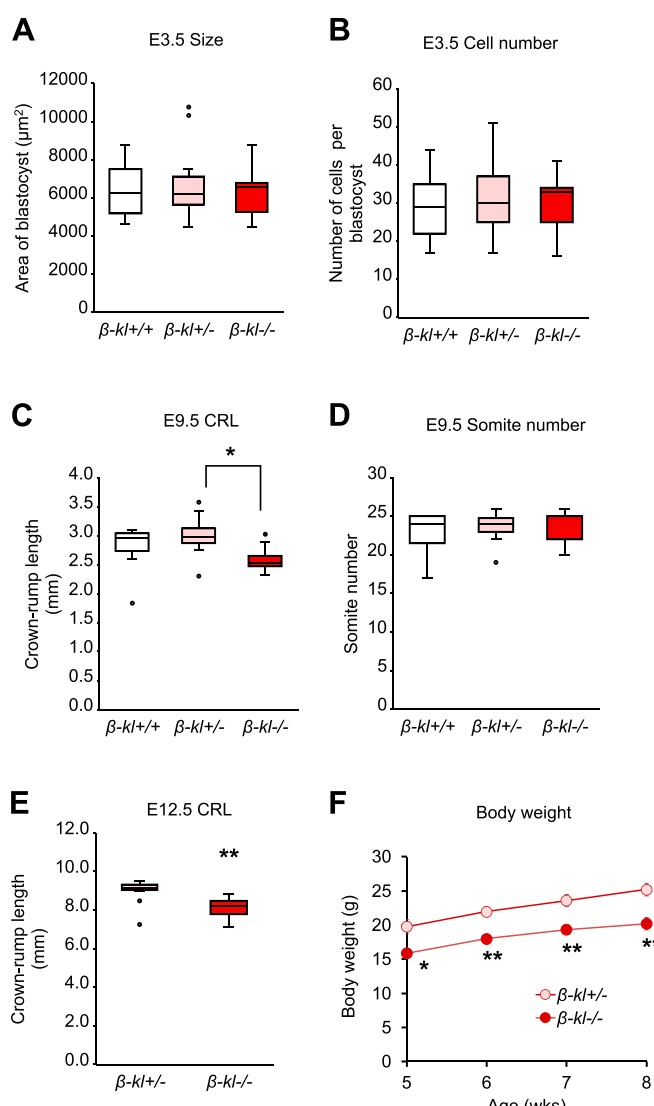

**Figure 1. β-KL is required for growth in postimplantation stages.**
**(A)** Size of β-kl+/+ (n = 13), β-kl+/− (n = 25), and β-kl−/− (n = 13) blastocysts at E3.5. **(B)** Total cell numbers of β-kl+/+ (n = 13), β-kl+/− (n = 25), and β-kl−/− (n = 13) blastocysts at E3.5. The same samples as in (A) were used. **(C)** Crown-rump length in β-kl+/+ (n = 9), β-kl+/− (n = 12), and β-kl−/− (n = 9) embryos at E9.5. **(D)** Somite numbers in β-kl+/+ (n = 9), β-kl+/− (n = 12), and β-kl−/− (n = 9) embryos at E9.5. The same samples as in (C) were used. **(E)** Crown-rump length in β-kl+/− (n = 10) and β-kl−/− (n = 13) embryos at E12.5. **(F)** Body weight changes in male β-kl+/− (n = 8) and β-kl−/− (n = 6) mice. The samples used in (A, B, C, D) were obtained from female β-kl+/− mice mated with male β-kl+/− mice. In (E), the samples were collected from female β-kl+/− mice mated with male β-kl−/− mice. Data information: in (A, B, C, D, E), midlines represent the median, boxes the interquartile range (25th to 75th percentile), and whiskers the range of data. *P < 0.05, **P < 0.01. **(A, B, C, D)** Mann–Whitney's U-test with the Bonferroni correction. **(E)** Mann–Whitney's U-test. In (F), data are presented as mean ± SEM. *P < 0.05, **P < 0.01 (t test). See also Fig S1.
Source data are available for this figure.

confirmed the presence of β-KL protein in the E9.5 yolk sac (Fig 3B). The molecular weight of β-KL in the yolk sac was slightly smaller than that of β-KL expressed in the adult liver, which may suggest that there is some difference in form or modifications of the protein

between adult and embryonic tissues. Hematoxylin and eosin staining revealed that the yolk sac from β-kl−/− embryos at E9.5 appeared grossly normal, but there were vacuoles in the baso-lateral surface (Fig 3C). Raabe et al reported that the vacuoles in yolk sacs lacking microsomal triglyceride transfer protein (Mttp) were cytosolic lipid droplets (Raabe et al, 1998). In accordance with this, we confirmed lipid accumulation in β-kl−/− yolk sacs by Oil Red O staining (Fig 3D).

Next, we analyzed gene expression in the yolk sac by qRT-PCR. Apolipoprotein B (Apob) and Mttp are essential for lipoprotein synthesis and their deletion causes lipid accumulation in the yolk sac (Farese et al, 1996; Raabe et al, 1998). However, the mRNA levels of Apob and Mttp were not altered in β-kl−/− yolk sacs (Fig 3E). Among the lipogenic genes, Hmgcs1, which converts acetyl-CoA to HMG-CoA in the mevalonate pathway, was up-regulated in β-kl−/− yolk sacs, but the expression of Hmgcr, the rate-limiting enzyme of de novo cholesterol synthesis, showed no difference between β-kl+/− and β-kl−/− yolk sacs (Fig 3E). Fasn mRNA levels were slightly but significantly increased in β-kl−/− yolk sacs (Fig 3E), whereas the levels of lipoprotein receptors (Ldlr, Scarb1) were not altered (Fig 3E). We previously demonstrated that the livers in β-kl−/− mice showed increased lipoprotein lipase (Lpl) mRNA levels, and the uptake of very low-density lipoprotein into the liver was increased (Kobayashi et al, 2016). Similarly, the expression levels of lipases (Lpl, Lipa, Lipc), digestive enzymes that are required for lipid uptake, were significantly increased in β-kl−/− yolk sacs (Fig 3E). Although the physiological significance of these enzymes in the yolk sac is unclear, we confirmed that these enzymes are detectable at the mRNA level in the yolk sac (Fig S4).

Because the murine embryo absorbs lipids derived from maternal blood through the yolk sac before placental maturation (Zohn & Sarkar, 2010), we speculated that β-KL expressed in the yolk sac might regulate the cholesterol supply from the dam to the embryo. To address this hypothesis, we injected $^{13}$C-labeled free cholesterol (FC) into pregnant mice at E8.5 and determined the $^{13}$C-cholesterol contents in the whole embryos after 24 h by GC-MS. As previously reported (Yoshida & Wada, 2005), $^{13}$C-FC was detected in the embryo (Fig 3F), confirming that maternal lipoproteins were transported to the embryo through the yolk sac. $^{13}$C-FC in β-kl−/− embryos was significantly decreased compared with β-kl+/− embryos (Fig 3F), indicating that β-KL is involved in feto-maternal cholesterol transport. Consistent with the TLC analyses, we found that $^{12}$C-FC (the sum of exogenously supplied and endogenously synthesized cholesterol) tended to be decreased in embryos lacking β-kl (Fig 3F).

To clarify the contribution of β-KL in the yolk sac to growth and lipid regulation, we analyzed β-kl−/−/Tg mice, which were generated by crossing β-kl−/− mice with hepatocyte-specific β-kl transgenic (Tg) mice (Kobayashi et al, 2016). As expected, the β-kl mRNA level in the β-kl−/−/Tg embryos was comparable with that in β-kl+/− embryos (Fig S5A). β-kl expression in β-kl−/−/Tg yolk sacs was very low (Fig S5B), confirming the loss of β-KL expression in yolk sacs of β-kl−/−/Tg mice. The CRL of the β-kl−/−/Tg embryos tended to be shorter than that of β-kl−/− embryos (Fig S5C). As in β-kl−/− embryos, Hmgcr and Ldlr mRNA levels were elevated in the β-kl−/−/Tg embryos (Fig S5D). The expression of Lpl in the β-kl−/−/Tg yolk sacs was increased, though not significantly (Fig

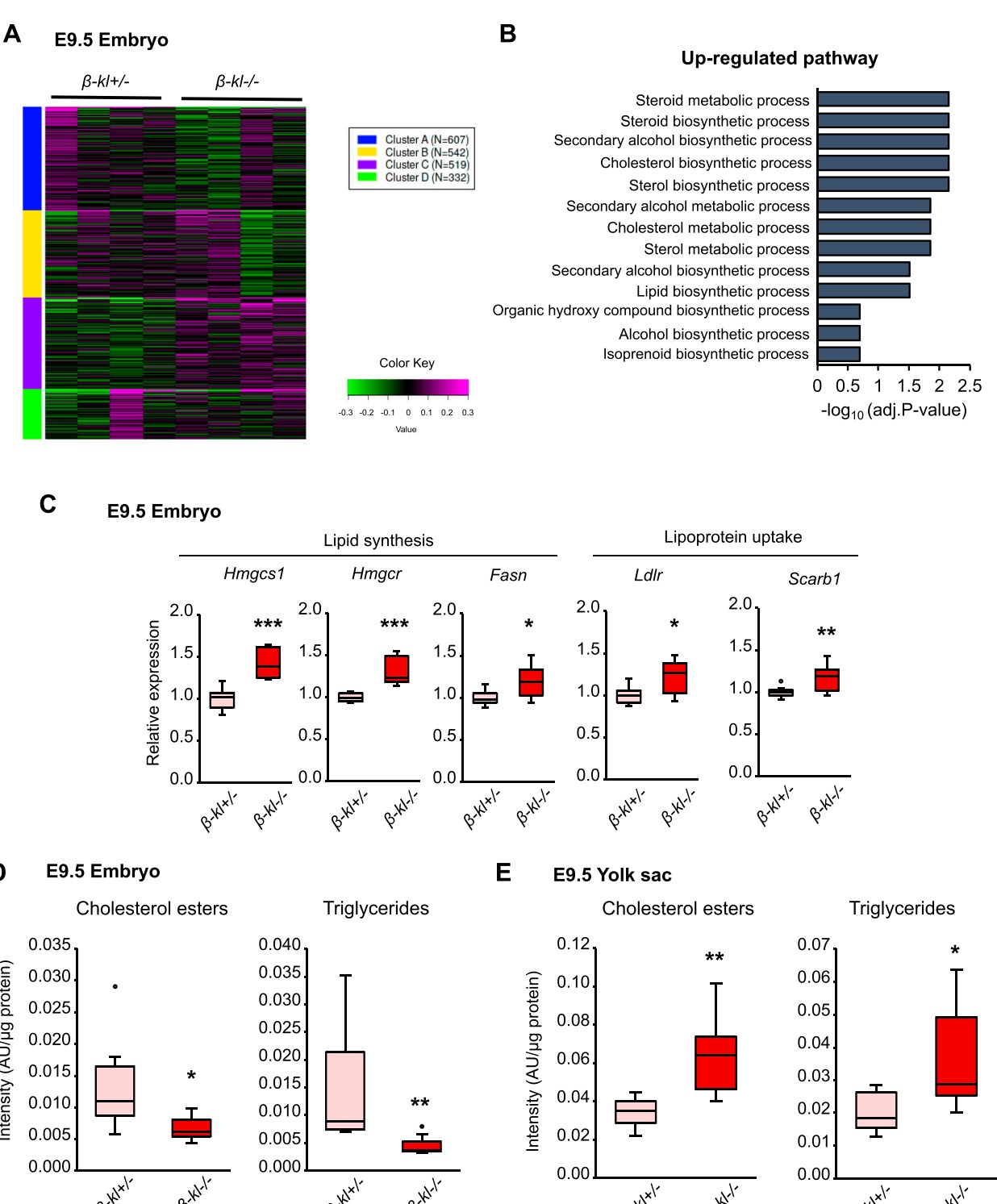

**Figure 2. β-KL deletion leads to lipid abnormalities in E9.5 embryo and yolk sac.**
**(A)** k-means clustering of the top 2,000 genes altered in β-kl−/− embryos at E9.5. **(B)** Pathways up-regulated in β-kl−/− embryos at E9.5 analyzed by the GAGE methods. **(C)** mRNA expression of genes involved in the regulation of lipid metabolism in the embryos at E9.5 (β-kl+/−, n = 11; β-kl−/−, n = 8). Data are shown as fold increase over the average expression levels in β-kl+/− embryo. **(D)** The levels of cholesteryl ester (CE) and triglycerides (TG) per μg of protein in the embryo at E9.5 measured by TLC (β-kl+/−, n = 8; β-kl−/−, n = 7). **(E)** The levels of CE and TG per μg of protein in the yolk sac at E9.5 measured by TLC (β-kl+/−, n = 7; β-kl−/−, n = 8). In (E), yolk sacs were obtained from the same animals used in (D). All samples were obtained from female β-kl+/− mice mated with male β-kl−/− mice. Data information: in (C, D, E), midlines represent the median, boxes the interquartile range (25th to 75th percentile), and whiskers the range of data. *P < 0.05, **P < 0.01, ***P < 0.001 (Mann–Whitney's U-test). See also Figs S2 and S3.
Source data are available for this figure.

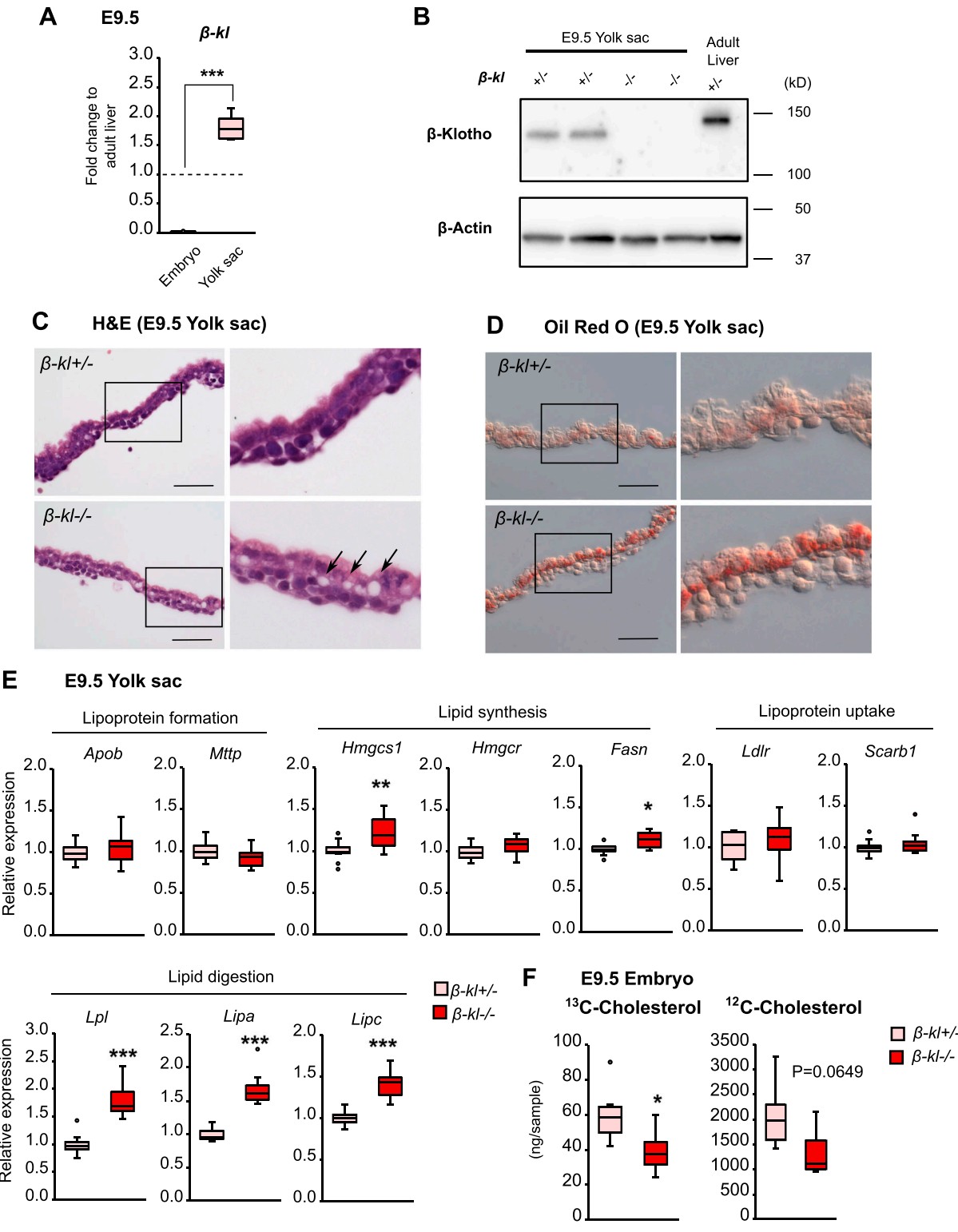

**Figure 3. β-KL is required to regulate maternal cholesterol transport through the yolk sac.**
**(A)** mRNA levels of *β-kl* in *β-kl+/−* embryos (n = 12) and yolk sacs (n = 11) at E9.5. The embryos and yolk sacs were prepared from the same animal. Data are shown as fold increase over the average expression levels in adult liver (n = 4). **(B)** Representative image of Western blot of *β-KL* protein in tissue lysate (5 μg of protein) of *β-kl+/−* and *β-kl−/−* yolk sacs at E9.5. Lysate of the adult liver (10 μg of protein) was loaded as a positive control. The liver was obtained from male *β-kl+/−* mice (12 wk of age). **(C)** Representative images of hematoxylin and eosin staining of yolk sacs from *β-kl+/−* and *β-kl−/−* embryos at E9.5. Arrowheads indicate vacuoles. Scale bars: 50 μm. Insets are enlargements of the boxes. **(D)** Representative images of Oil Red O staining of yolk sacs from *β-kl+/−* and *β-kl−/−* embryos at E9.5. Scale bars: 50 μm. Insets are enlargements of the boxes. **(E)** mRNA expression of genes involved in the regulation of lipid metabolism in the yolk sac at E9.5 (n = 11 each). Data are shown as fold increase

S5E). We evaluated the LDL uptake in the yolk sac in an ex vivo experiment (Fig S5F), because cholesterol-rich lipoproteins, such as LDL, are absorbed via receptor-mediated endocytosis. At E9.5, the fluorescence of BODIPY-labeled LDL was observed in the β-kl+/− yolk sac after 30 min of incubation (Fig S5G). BODIPY signals were also detected in β-kl−/− and β-kl−/−/Tg yolk sacs, showing that LDL uptake occurred in the absence of β-KL. The dye emission in β-kl−/− yolk sac shifted from green to red, which indicates the accumulation of BODIPY molecules. In tissue sections prepared from the yolk sacs incorporating BODIY-LDL, BODIPY-labeled vesicles were spread throughout cells in the β-kl+/−/Tg yolk sac. The vesicles were mainly localized close to the nucleus and seemed larger in the β-kl−/−/Tg yolk sac than in the β-kl+/−/Tg yolk sac (Fig S5H). Taken together, these results suggest that β-KL expressed in yolk sac is required for lipid transport through the yolk sac.

### β-KL–FGF15 axis regulates growth and lipid metabolism in the embryonic stages

Hepatic β-KL regulates lipid and bile acid synthesis by mediating FGF15 signaling from the ileum in adult mice (Tomiyama et al, 2010; Kobayashi et al, 2016). Because Fgf15 mRNA is expressed in the developing central nervous system in mice (McWhirter et al, 1997; Gimeno et al, 2002), we examined whether β-KL interacts with FGF15 in the embryonic stages.

In contrast to β-kl, the Fgf15 mRNA level in E9.5 embryos was ~170-fold higher than that in the yolk sac (Fig 4A). Notably, the Fgf15 level in E9.5 embryos was 1.6-fold higher than that in the adult ileum (Fig S6A). Our previous studies showed that Fgf15 mRNA expression in the ileum was elevated in adult β-kl−/− mice (Tomiyama et al, 2010; Kobayashi et al, 2016). Similarly, the level of Fgf15 mRNA in β-kl−/− embryos was significantly higher than that in their β-kl+/− littermates (Fig 4A).

Activation of FGF signaling induces ERK phosphorylation, which results in modulation of the expression levels of the downstream genes (Tomiyama et al, 2010). Dual-specific phosphatases (DUSPs) are a subclass of protein tyrosine phosphatases that promote a negative feedback loop to specifically dephosphorylate ERKs (Jeffrey et al, 2007). The genes encoding DUSP are targets of ERK, and therefore, their mRNA levels reflect ERK activity (Ekerot et al, 2008; Lan et al, 2017). To examine whether β-KL is required to transduce FGF signaling in embryonic tissues, we measured mRNA levels of DUSPs. At E9.5, Dusp4 and Dusp6 mRNA levels were decreased in β-kl−/− yolk sacs, but not in β-kl−/− embryos (Fig 4B). These data suggest that FGF15 is mainly produced in the embryo and activates ERK1/2 signaling through β-KL expressed in the yolk sac.

Next, we analyzed Fgf15−/− embryos and yolk sacs at E9.5 to verify the contribution of FGF15 to growth and lipid regulation. The CRL of Fgf15−/− embryos was significantly smaller than that of their WT or heterozygous littermates (Fig 5A), confirming that FGF15 is required

for fetal growth. Fgf15−/− embryos displayed increased expression levels of genes involved in lipid synthesis and lipoprotein uptake (Fig 5B), in accordance with the pattern of expression in β-kl−/− embryos. The expression profile of genes involved in lipid metabolism in Fgf15−/− yolk sacs was consistent with that in β-kl−/− yolk sacs (Fig 5C). Oil Red O staining showed lipid accumulation in Fgf15−/− yolk sacs (Fig 5D). The levels of Dusp4 and Dusp6 mRNAs were down-regulated in Fgf15−/− yolk sacs (Fig 5E). The expression of Dusp4 and Dusp6 mRNA was significantly up-regulated in Fgf15−/− embryos, unlike β-kl−/− embryos. These data supported the idea that ERK1/2 signaling is activated in the yolk sac via the β-KL–FGF15 interaction.

In this study, we mainly used β-kl+/− embryos as a control. To examine whether one allele of β-kl was enough to suppress Fgf15 expression, we measured the Fgf15 levels in β-kl+/+ embryos and yolk sacs. β-kl mRNA was expressed dose-dependently in β-kl+/+ and β-kl+/− tissues (Fig S6B and C), whereas the Fgf15 mRNA levels were similar in β-kl+/+ and β-kl+/− embryos (Fig S6D). We also compared β-kl expression in Fgf15 mutant lines. Fgf15 mRNA was expressed dose-dependently in Fgf15+/+ and Fgf15+/− embryos (Fig S6E), but there was no difference in β-kl expression levels in the yolk sac among the genotypes (Fig S6F). These data show that one allele of β-kl is sufficient to regulate Fgf15 expression.

We next examined the contribution of FGF21, another partner of β-KL (Owen et al, 2015). Unlike Fgf15, the normalized read counts of Fgf21 were extremely low in E9.5 embryos, and no difference was observed between β-kl+/− and β-kl−/− embryos (Fig S6G).

Finally, we examined the levels of FGFRs in the yolk sac and embryo (Fig S6H). In adult mice, β-KL serves as a coreceptor of FGFR4 in the liver (Tomiyama et al, 2010). Among the four subtypes, Fgfr1 and Fgfr2 mRNAs showed significantly higher levels in embryos than in yolk sacs. Fgfr3 mRNA was expressed at a slightly, but significantly higher level in yolk sacs than in embryos. The Fgfr4 mRNA level in yolk sacs was 15-fold higher than that in embryos, suggesting that FGFR4 in the yolk sac is required to receive the FGF15 signal from the embryo.

## Discussion

β-KL plays critical roles in whole-body metabolism by mediating FGF15 and 21 signaling in adult mice (Ito et al, 2005; Tomiyama et al, 2010; Ding et al, 2012; Bookout et al, 2013; Owen et al, 2013; Kobayashi et al, 2016; Coate et al, 2017). A strong expression of β-kl mRNA has been detected not only in adult tissues, but also in embryonic tissues (Ito et al, 2000); however, the developmental role of β-KL has remained unknown. Thus, the aim of this study was to determine the role of β-KL in the developmental stages. Specifically, we hypothesized that β-KL expressed in the yolk sac, the predominant site of β-kl expression in fetal tissues, is required for

---

over the average expression levels in β-kl+/− yolk sac. **(F)** The levels of $^{13}$C-labeled and native ($^{12}$C) free cholesterol in the whole embryos at E9.5 were measured by GC-MS (n = 6 each). All samples were collected from female β-kl+/− mice mated with male β-kl−/− mice. Data information: in (A, E, F), midlines represent the median, boxes the interquartile range (25th to 75th percentile), and whiskers the range of data. *P < 0.05, **P < 0.01, ***P < 0.001 (Mann–Whitney's U-test). See also Figs S4 and S5. Source data are available for this figure.

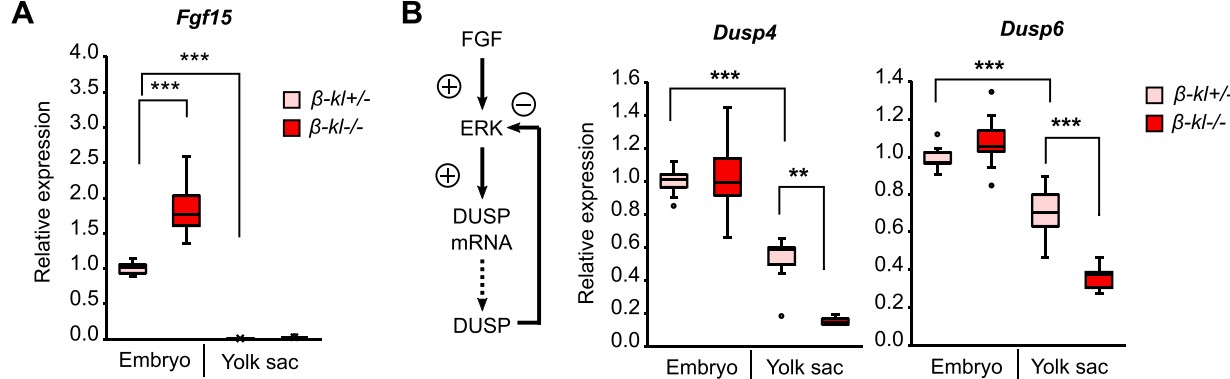

**Figure 4.  β-KL is required to activate ERK signaling in the yolk sac.**
**(A)** mRNA levels of *Fgf15* in the embryo (*β-kl+/−*, n = 11; *β-kl−/−*, n = 12) and yolk sac (n = 11 each) at E9.5. **(B)** mRNA levels of *Dusp4* and *Dusp6* in the embryo (*β-kl+/−*, n = 11; *β-kl−/−*, n = 12) and yolk sac (*β-kl+/−*, n = 9; *β-kl−/−*, n = 10) at E9.5. Data are shown as fold increase over the average expression levels in *β-kl+/−* embryos. In (A, B), the same amount of RNA was used for each sample. The samples were obtained from female *β-kl+/−* mice mated with male *β-kl−/−* mice. Data information: in (A, B), midlines represent the median, boxes the interquartile range (25th to 75th percentile), and whiskers the range of data. **$P < 0.01$, ***$P < 0.001$ (Mann–Whitney's $U$-test with the Bonferroni correction).
Source data are available for this figure.

metabolic regulation in the embryo, consequently controlling fetal growth and postnatal body size.

This hypothesis is supported by the following lines of evidence. First, *β-kl−/−* embryos were indistinguishable from WT littermates at the blastocyst stage but were significantly smaller in the post-implantation stages, in which the yolk sac contributes to fetal growth. β-KL expression was observed in both the embryo and yolk sac, but the transgenic rescue of β-KL expression in embryos failed to restore growth restriction, suggesting that β-KL expressed in the yolk sac is required for fetal growth. Second, *β-kl−/−* embryos exhibited a marked reduction of CE and TG contents, concomitantly with growth restriction. In contrast, lipid accumulation was observed in the yolk sac from *β-kl−/−* embryos. Stable isotope analysis showed that maternal blood-derived cholesterol levels were reduced in *β-kl−/−* embryos, demonstrating that β-KL is required to supply maternal cholesterol to embryos. Moreover, we found that deletion of *Fgf15* led to similar growth and lipid abnormalities in the embryo and yolk sac. ERK activation, downstream of FGF signaling, was reduced in yolk sacs of both *β-kl−/−* and *Fgf15−/−* mice, but not in the embryos, which suggests that the yolk sac responds to the FGF15 signal. Because *β-kl* is predominantly expressed in yolk sac, whereas *Fgf15* expression in the embryo is much higher than that in the yolk sac, we consider that FGF15 secreted from the embryo is transported to the yolk sac and its signal is transduced in cooperation with β-KL in the yolk sac. The inter-tissue communication between hepatic β-KL and ileal FGF15 for the regulation of whole-body lipid metabolism is well established, but the role of the β-KL-FGF15 axis in embryonic development has not been reported. Thus, we have uncovered a novel function of the β-KL-FGF15 axis, promoting fetal growth through the regulation of feto-maternal cholesterol transport.

Interestingly, Yuan reported that epigenetic modulation of FGF21 activity during the late-gestation stages affects lipid metabolism in postnatal mice (Yuan et al, 2018). However, we found that *Fgf21* expression was almost undetectable at E9.5. Although the possibility that β-KL interacts with FGF21 in the embryonic stages cannot

be excluded, our results suggest that β-KL mainly interacts with FGF15 at E9.5.

The rodent embryo itself synthesizes cholesterol for normal development but also uses cholesterol derived from maternal blood (Yoshida & Wada, 2005). Although defects in de novo cholesterol synthesis can cause severe congenital birth abnormalities (such as Smith–Lemli–Opitz syndrome), it remains controversial whether cholesterol derived from maternal blood is involved in organogenesis or growth in embryos. Considering that *β-kl−/−* and *Fgf15−/−* embryos are morphologically normal but smaller than the control, it is reasonable to suppose that maternal cholesterol contributes to fetal growth rather than organogenesis. Cholesterol serves as a component of cell membranes, a precursor of hormones and metabolic mediators such as oxysterols, and an activator of sonic hedgehog signaling (Woollett, 2005, Blassberg & Jacob, 2017). Thus, it is intuitively understandable that a deficiency of cholesterol would inhibit the growth of embryos, though further studies will be needed to clarify the precise mechanisms.

The smaller body size of *β-kl−/−* embryos persists after birth, indicating that the lower body weight in postnatal *β-kl−/−* mice results from FGR. It is well-known that low birth weight is inversely related to midlife dysmetabolism (Barker, 2007; Hsu et al, 2021). Especially, infants who were born small and experienced rapid infantile growth or "catch-up growth" have a higher risk of cardiovascular disease, type 2 diabetes, and dyslipidemia in adulthood (Leunissen et al, 2009; Singhal, 2017). Therefore, the long-term adverse effects of catch-up growth are of considerable interest, but the underlying mechanisms are poorly understood. Interestingly, the lower body weight in *β-kl−/−* mice was not ameliorated even under high-fat diet conditions (Kobayashi et al, 2016; Somm et al, 2017). Thus, *β-kl−/−* mice could be a useful model for investigating the mechanism of catch-up growth.

In the present study, no sex bias was observed in the severity of FGR in *β-kl−/−* embryos. Furthermore, reduced body weight in postnatal *β-kl−/−* mice was seen similarly in females and males (Ito et al, 2005). These data suggest that the contribution of β-KL to

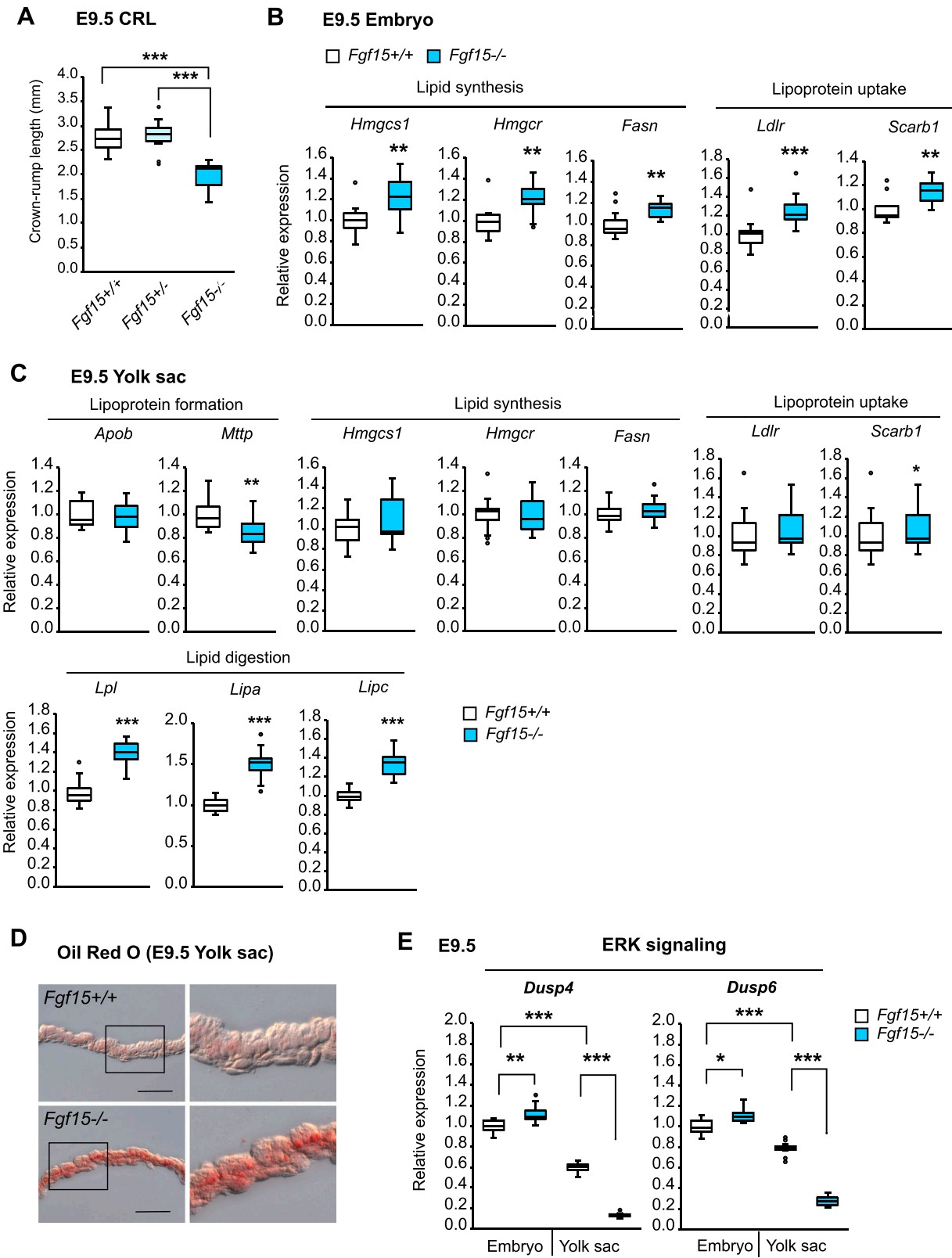

**Figure 5.  β-KL–FGF15 axis regulates growth and lipid metabolism in the embryonic stages.**
**(A)** Crown-rump length in *Fgf15+/+* (n = 10), *Fgf15+/-* (n = 15), and *Fgf15-/-* (n = 7) embryos at E9.5. **(B)** mRNA expression of genes involved in the regulation of lipid metabolism in embryos at E9.5 (*Fgf15+/+*, n = 12–14; *Fgf15-/-*, n = 14–15). Data are shown as fold increase over the average expression levels in *Fgf15+/+* embryos. **(C)** mRNA expression of genes involved in the regulation of lipid metabolism in the yolk sac at E9.5 (*Fgf15+/+*, n = 14; *Fgf15-/-*, n = 13–15). Data are shown as fold increase over the

growth is independent of gender. On the other hand, phenotypic assessments of β-kl−/− mouse metabolism have mainly been done using male mice. In this context, the sex differences in the long-term consequences of FGR have been reported both in animal models and humans (Eriksson et al, 2010; Sohi et al, 2011; Cheong et al, 2016). Thus, we need to include female mice in future studies to determine the metabolic outcomes of FGR in β-kl−/− mice.

Our study has uncovered a critical role of the yolk sac in the onset of FGR in mice. In humans, the yolk sac attaches to the conceptus like a balloon, whereas in rodents, the embryo is surrounded by the yolk sac and floats within it. But, regardless of the structural differences among species, the role of the yolk sac in exporting maternal nutrients to the embryo is the same (Zohn & Sarkar, 2010). Although the importance of the yolk sac in fetal growth is controversial in humans, transcriptome analysis has revealed that pathways contributing to lipid transport are conserved not only in mice, but also in humans (Cindrova-Davies et al, 2017). The conservation of transcriptomes between humans and mice suggests that the human yolk sac is also likely to contribute to human fetal growth. Whether our findings are directly applicable to human yolk sacs is uncertain, but β-KL and FGF19 (the human ortholog of FGF15) regulate bile acid synthesis and plasma lipid levels in humans in a similar manner (Lundåsen et al, 2006). Thus, it seems plausible that the β-KL–FGF19 axis would operate in human embryos, although further investigation will be needed to confirm this.

In conclusion, we have identified a novel regulatory mechanism of fetal growth and feto-maternal cholesterol transport, mediated by the β-KL–FGF15 axis in mice. The pathological mechanisms underlying FGR are complex, but our present findings might be helpful in developing strategies for preventing FGR and subsequent metabolic disorders in later life.

# Materials and Methods

## Animals

β-kl−/− mice, Fgf15−/− mice, and hepatocyte-specific β-kl transgenic mice were generated as previously described (Wright et al, 2004; Ito et al, 2005; Kobayashi et al, 2016). β-kl−/− mice were maintained on a mixed background. Hepatocyte-specific β-kl transgenic mice were identified by using forward primer 5′-CAT ATA AAT TCT GGC TGG CGT G-3′ and reverse primer 5′-GTT GAC ACC TCT CAG GTG TGA GT-3′. The sex of the embryos was determined by using primers for Zfy (forward: 5′-CCT ATT GCA TGG ACT GCA GCT TAT G-3′, reverse: 5′-GAC TAG ACA TGT CTT AAC ATC TGT CC-3′) (Bradford et al,

2009). C57BL/6J mice were purchased from CLEA Japan. To obtain embryos, one male mouse was allowed to cohabit with one or two female mice overnight. Noon on the day of detection of the vaginal plug was considered as E0.5. The genotype of the mated mice is indicated in the figure legends. Mice were maintained on a 12-h dark–light cycle and given free access to food and water. All animals were euthanized by cervical dislocation or isoflurane inhalation. All experiments were approved by RIKEN Center for Biosystems Dynamics Research, Foundation for Biomedical Research and Innovation at Kobe, and Kyoto University.

## Sample preparation

Based on the milestones of intrauterine development in mice (Cross et al, 1994; Rossant & Cross, 2001), tissues were collected at E3.5, E9.5, and E12.5, corresponding to preimplantation, yolk sac-dependent, and placenta-dependent stages, respectively. Blastocysts were obtained at E3.5 as previously described (Nagy et al, 2002). After microscopic observation, the blastocysts were lysed and used as a template for genotyping PCR (Honda et al, 2019). Whole embryos and yolk sacs were collected at E9.5 under microscopic observation. Fetal livers, yolk sacs, and placentas were dissected at E12.5. To avoid contamination with maternal tissues or blood, the specimens were washed with PBS. A portion of the yolk sac, allantois or embryo was used for genotyping. Adult livers and small intestines were collected from male C57BL/6J mice (10 wk of age) in a fed state. The small intestine was divided into three parts (0%, 33%, 66%, and 100%; relative distance from the stomach to distal ileum) and a piece of tissue collected from the middle of the 66–100% part was used as the ileum. Each sample was prepared from a different animal, unless mentioned in the figure legends.

## RNA-sequencing analysis

Total RNA was extracted from E9.5 embryos using an RNeasy Mini kit (QIAGEN) and treated with DNase I (QIAGEN) on-column according to the manufacturer's instructions. The quality of RNA was confirmed by using an Experion Automated Electrophoresis System (Bio-Rad). Sequencing was performed at 100 bp paired-ends on an Illumina HiSeq 2500 with all eight libraries multiplexed in one lane. Base-calling was processed with RTA 1.18.64 (Real-Time Analysis, HiSeq Control Software). Fastq files were generated with bcl2fastq 1.8.4 (Illumina). The quality of the RNA-Seq reads was evaluated using trimmomatic (ver. 0.32). Having established the high quality of the data, sequence reads for each library were mapped using BWA (ver. 0.7.15). Next, for each library, we estimated the number of sequence reads overlapping at any given nucleotide position in the reference genome at 100-bp resolution. The expression data were quantified using samtools version 0.1.19. Differentially expressed

average expression levels in Fgf15+/+ yolk sacs. (D) Representative images of Oil Red O staining of yolk sac from Fgf15+/+ and Fgf15−/− embryos at E9.5. Scale bars: 50 μm. Insets are enlargements of the boxes. (E) mRNA levels of Dusp4 and Dusp6 in the embryo and yolk sac at E9.5 (Fgf15+/+, n = 14; Fgf15−/−, n = 10). The same amount of RNA was used for each sample. Data are shown as fold increase over the average expression levels in Fgf15+/+ embryos. In (A, B, C, D, E), the samples were obtained from female Fgf15+/− mice mated with male Fgf15+/− mice. Data information: in (A, B, C, E), midlines represent the median, boxes the interquartile range (25th to 75th percentile), and whiskers the range of data. *P < 0.05, **P < 0.01, ***P < 0.001 (A, E) Mann–Whitney's U-test with the Bonferroni correction. (B, C) Mann–Whitney's U-test. See also Fig S6.
Source data are available for this figure.

genes were analyzed using integrated Differential Expression and Pathway analysis (iDEP ver. 0.93) (Ge et al, 2018; Ge, 2021).

## qRT-PCR analysis

Total RNA was extracted from whole embryo (E9.5), yolk sac (E9.5 or E12.5), liver (E12.5) or placenta (E12.5) using an RNeasy Mini Kit (QIAGEN) or AllPrep DNA/RNA/Protein Mini Kit (QIAGEN). Total RNA treated with DNase I (QIAGEN) on-column was transcribed into first-strand cDNA using a SuperScript VILO Master Mix (Thermo Fisher Scientific) according to the manufacturer's instructions. qRT-PCR analysis was performed by using TaqMan Expression Assay Primers (Applied Biosystems, Table S4). Expression levels were calculated by the relative standard curve method with 18S rRNA as an internal control and the data were shown as relative expression compared with the control group. In the comparison between fetal tissues and adult tissues, data were shown as fold increase over the average expression levels in the tissues of male C57BL/6J mice.

## Western blotting

Tissue homogenates of yolk sac (E9.5) and adult liver (male $\beta$-kl+/− mice, 12 wk of age) were prepared in buffer (20 mM 4-[2-hydroxyethyl]-1-piperazineethanesulfonic acid, 150 mM NaCl, and 0.5% Nonidet P-40 [pH 7.4]) containing protease inhibitor cocktail (cOmplete, Mini, EDTA-free; Roche), and phosphatase inhibitor cocktail (Nacalai tesque). The supernatant was electrophoresed on 4–20% gradient SDS-polyacrylamide gel and transferred to PVDF membranes. After blocking with 5% skim milk in TBS-T for 1 h at room temperature, the membrane was incubated overnight with goat anti-mouse KLB polyclonal antibody (AF2619, 1:1,000; R&D Systems) at 4°C. Then, the membrane was incubated with anti-goat HRP (705-035-147, 1:100,000; Jackson) for 45 min at room temperature. The signals were developed by using the ECL Prime Western Blotting Detection Reagents (Cytiva) and detected with a FUSION SYSTEM (Vilber-Lourmat). After stripping, the membrane was reprobed with rabbit anti-$\beta$-actin monoclonal antibody (4970, 1:1,000; Cell Signaling) and anti-rabbit HRP (A16110, 1:200,000; Thermo Fisher Scientific).

## TLC

Whole embryo and yolk sac without allantois were collected at E9.5. Allantoises were used for genotyping. Placentas were dissected at E12.5. Tissues were homogenized in 880 $\mu$l of hexane/isopropanol (3:2, vol/vol) by using zirconia beads (Sarstadt) in a Micro Smash (TOMY) at 4°C. The homogenate was centrifuged at 10,000$g$ for 5 min at 4°C. Lipids were extracted from the supernatants as previously described (Hara & Radin, 1978; Matsuoka et al, 2001). The pellets were solubilized with 2% SDS solution, and protein concentrations were determined by the BCA method. Extracted lipids were evaporated and resolubilized in chloroform, then applied to HPTLC plates (Merck). TLC nonpolar Lipid Mixture B (Matreya) was loaded as a loading control. Free cholesterol solubilized in ethanol (20 $\mu$g/plate) was used as an internal standard. The plates were developed in hexane/diethyl ether/acetic acid (100:15:1, v:v:v) at room temperature. Lipids were stained with Coomassie brilliant

blue (Nakamura & Honda, 1984). Spot images were obtained by scanning with a FUSION SYSTEM (Vilber-Lourmat). The signal intensity corresponding to cholesteryl ester and triglyceride in each sample was quantified using ImageJ's Fiji software (Schindelin et al, 2012), and normalized by the signal intensity of the internal standard. Signals were divided by protein concentration and expressed as normalized relative intensity per $\mu$g of protein.

## Tracer experiment

One milligram of [2,3,4–$^{13}$C]-cholesterol (Cambridge Isotope Laboratories) dissolved in 40 $\mu$l of ethanol was mixed with 10% Intralipos (Otsuka Pharmaceutical) (1:1, vol/vol) just before administration, and the lipid solution was intraperitoneally injected into pregnant $\beta$-kl+/− mice at E8.5. Then, the whole embryos were collected at E9.5 under microscopic observation. The yolk sacs and allantoises were removed. The collected embryos were homogenized in 1.1 ml of hexane and isopropanol (3:2, v:v), and the homogenate was centrifuged at 15,000$g$. The resulting supernatant (1.0 ml) was collected and stored at −80°C until use. Free cholesterol contents in each whole embryo with/without [2,3,4–$^{13}$C] labeling were determined by GC-MS (GCMS-QP2010Ultra; Shimadzu) as previously described (Yoshida & Wada, 2005). Cholesterol (D7) (Avanti Polar Lipids) was added to the sample tube as an internal standard. The content of free cholesterol in the tissue was calculated using standard curves generated by using unlabeled cholesterol or [2,3,4–$^{13}$C]-cholesterol (Cambridge Isotope Laboratories).

## Histologic analysis

Yolk sacs (E9.5) were fixed in 10% formalin (Wako) and embedded in paraffin. The paraffin sections were stained with hematoxylin and eosin. For Oil Red O staining, yolk sacs (E9.5) were fixed in 4% PFA for 1 h at 4°C, left on a 30% sucrose solution for 1 h at 4°C, and frozen in Tissue-Tek optimum cutting temperature compound (Sakura) for cryo-sectioning. Sections (14 $\mu$m) were mounted on glass slides (MATSUNAMI). Oil Red O (0.3%) in isopropanol solution was freshly prepared, mixed with $H_2O$ (0.3% Oil Red O isopropanol: $H_2O$ 3:2), and filtered through filter paper (Whatman). Sections were incubated in 60% isopropanol for 1 min and then in Oil Red O solution for 15 min and 60% isopropanol for 1 min. After rinsing in water, the sections were mounted in mounting media (Fluoromount-G; Thermo Fisher Scientific).

## LDL uptake assay

E9.5 embryo with yolk sac was dissected in PBS and incubated in 75% lipoprotein-deficient fetal calf serum (Sigma-Aldrich) and 25% DMEM for 30 min at 37°C in 5% $CO_2$. BODIPY FL-labeled LDL (BODIPY-LDL) (Thermo Fisher Scientific) was added at 40 $\mu$g/ml for 30 min at 37°C. After uptake, the embryo with yolk sac was washed in ice-cold PBS (+) three times to remove surface-bound BODIPY-LDL. The yolk sac was separated from the embryo and fixed in 4% PFA for 1 h at 4°C. The embryo was used for genotyping PCR. Fixed yolk sac was embedded in an optimum cutting temperature compound and sectioned by a cryostat. Fourteen $\mu$m cryostat sections were rehydrated in PBS and permeabilized in 0.1% Triton X-100 in PBS for

20 min at room temperature. Tissues were washed in PBS containing 0.1% Tween 20 (PBST) 3 times, and incubated with 4′,6-diamidino-2-phenylindole (DAPI, 1:1,000) (Dojindo) for 15 min. Sections were washed in PBS three times and mounted using Fluoromount-G (Thermo Fisher Scientific). Fluorescence images were obtained by Nikon A1R (Nikon). The color, brightness, and contrast of the images were adjusted using ImageJ's Fiji software.

## Statistics

All statistical analyses were performed with EZR (version 1.52) (Kanda, 2013). Statistical analyses were performed using a two-tailed $t$ test or Mann–Whitney's $U$-test (two groups) as indicated in figure legends. Bonferroni correction was applied for multiple-group analysis (three or more groups). $P < 0.05$ was considered significant. The number of replicates is given in each figure legend.

## Data Availability

The RNA-sequencing data from this publication have been deposited to the DDBJ database https://www.ddbj.nig.ac.jp/index.html and assigned the identifier DRA014754.

## Supplementary Information

## Acknowledgements

We thank Dr. Yasuhide Furuta (RIKEN Center for Biosystems Dynamics Research) and Dr. Cornelis Murre (University of California, San Diego) for providing *Fgf15−/−* mice. This project was supported by grants from Japan Agency for Medical Research and Development under Grant Number JP21gm5010002 (T Kawauchi), and KAKENHI from Japan Society for the Promotion of Science under Grant Numbers JP 26870926 (K Kobayashi) and JP 16K21714 (K Kobayashi).

## Author Contributions

K Kobayashi: conceptualization, data curation, formal analysis, funding acquisition, investigation, visualization, methodology, project administration, and writing—original draft, review, and editing.
K Iwasa: investigation, methodology, and writing—review and editing.
R Azuma-Suzuki: formal analysis, investigation, methodology, and writing—review and editing.
T Kawauchi: conceptualization, supervision, funding acquisition, and writing—original draft, review, and editing.
Y-i Nabeshima: conceptualization, resources, supervision, project administration, and writing—original draft, review, and editing.

## Conflict of Interest Statement

The authors declare that they have no conflict of interest.

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
