## [Reviewer comments · Life Science Alliance]

Life Science Alliance

Feto-maternal cholesterol transport regulated by β -Klotho-FGF15 axis is essential for fetal growth

Kanako Kobayashi, Kazuko Iwasa, Rika Azuma-Suzuki, Takeshi Kawauchi, and Yo-ichi Nabeshima

DOI:<https://doi.org/10.26508/lsa.202301916>

Corresponding author(s): Kanako Kobayashi, Kyoto University and Yo-ichi Nabeshima, Kyoto University

Review Timeline:

Submission Date:	2023-01-10
Editorial Decision:	2023-03-20
Revision Received:	2023-06-18
Editorial Decision:	2023-07-16
Revision Received:	2023-07-24
Accepted:	2023-07-26

Scientific Editor: Novella Guidi

Transaction Report:

March 20, 2023

Re: Life Science Alliance manuscript #LSA-2023-01916-T

Kanako Kobayashi
Kyoto University Graduate School of Medicine

Dear Dr. Kobayashi,

Thank you for submitting your manuscript entitled "Yolk sac-mediated feto-maternal cholesterol transport regulated by β -Klotho-FGF15 axis is essential for fetal growth" to Life Science Alliance. The manuscript was assessed by expert reviewers, whose comments are appended to this letter. We invite you to submit a revised manuscript addressing the Reviewer comments.

Thank you for this interesting contribution to Life Science Alliance. We are looking forward to receiving your revised manuscript.

Sincerely,

B. MANUSCRIPT ORGANIZATION AND FORMATTING:

Reviewer #1 (Comments to the Authors (Required)):

The authors here focus on a mouse model for fetal growth restriction (FGR). Fetal growth restriction is a complication of pregnancy with lasting effects for the offspring, clearly warranting research in this area. The authors use a genetic model where beta Klotho (b-KL) has been deleted. The b-kl-ko mice had already previously been shown to be growth restricted. B-KL is known to be expressed in the placenta at low levels, but in the yolk sac at high levels. The authors nicely demonstrate that FGR here occurs in the first half of pregnancy (when the yolk sac is more active than the placenta) and that cholesterol supply is also impaired.

This manuscript is clearly written and follows a logical strategy to characterize the phenotype. However, I have several concerns:

- In the human clinical situation, there are differences between boys and girls regarding the severity of FGR as well as the long-term outcome. Can the authors give some details about sex differences in their study? For example, was the ratio male/female the same in all genotypes, and how does the growth curve of the adolescent mice (Fig 1a) look like when split by sex?
- Fetal and yolk sac gene expressions are interesting, but what do they mean? Maybe the changes in expression in Fig 2-4 occur on a very low overall level? Can the authors show how high these expressions are compared to adult liver, for example? To judge the meaning of changes for example in bile acid metabolism (Cyp7a1) this would be helpful, as to the best of my knowledge bile acids are not produced that early in life.
- On page 6 the authors speculate about compensatory lipid synthesis, based on the TLC data. They should also consider the size of the yolk sac vs the embryo, as this might influence their conclusions. Also the placenta could be used as a control, because if the authors' hypothesis is correct it should show no differences.
- It is indeed fascinating that this very early event has such long-lasting consequences. However, the discussion goes too far by saying "... suggesting that cholesterol shortage during the embryonic period has a lifelong impact on the body size." as the authors cannot exclude that other molecules mediate this effect (for example oxysterols or bile acids).

Minor concerns:

- In the introduction, the statements about long term consequences of FGR (Barker hypothesis) and FGR itself are a bit mixed, this could be better structured. I would also prefer to have clearly defined aims rather than indirectly mentioned ones.
- A vaginal plug was used as a sign of successful mating - does this mean that the males were with the females for more than one night? How can you make sure that timing is precise?

Reviewer #2 (Comments to the Authors (Required)):

This manuscript by K. Kobayashi et al. attempts to demonstrate that the B-KL/FGF axis regulates feto-maternal cholesterol transport through the yolk sac, and that its dysfunction leads to FGR in mice. The study uses a B-KL knockout and a range of methods to study the underlying mechanisms explaining the link between altered B-KL function and fetal growth. Overall this study could demonstrate important findings to better understand FGR mechanistically. However, in its current form the study lacks an explicitly stated central hypothesis, the Methods are incomplete (and experiments therefore not reproducible), the Results read more like a combination of Methods and Discussion, and the Discussion itself is underdeveloped. There are tons of figures and a robust Supplemental Figure library that at times is confusing to toggle to from the general manuscript and a host of simple errors in the consistency between Figures/Figure legend. All of these nuances made it difficult to focus on the underlying science in order to completely comment on the impact the study results might have on the field. Major revisions are needed in order for another review to ascertain readiness for publication.

MAJOR POINTS

1- Lack of clarity and/or reproducible detail

a. Hypothesis/study goal. Neither of these are explicitly stated in either the Abstract or Introduction. The closest the authors come to stating a hypothesis is found on page 8 in Results subsection Deletion of B-KL leads to lipid abnormalities in the yolk

sac, with the sentence starting with "Since the embryo absorbs cholesterol..." A hypothesis really should be explicitly stated earlier in the paper to provide the framework for all other sections, and should be re-visited in the Discussion section.

b. Methods. There is much detail lacking in this section that would help reviewers and readers. Experiments should be reproducible, but there is not enough detail or references provided for this to be possible. See specific examples below:

i. Animals-

1. No mention of the type of mouse used, until it is mentioned in Results supplementary Fig S1A legend (C57BL6/J).
2. No mention of the embryonic day time points at which animals were sacrificed/tissues collected for the various experiments. We learn in the results section what these time points are, but should be included in Methods, along with some mention of the significance of selected time points (ie. E9.5 and E12.5).

ii. Specific tissue types for each experiment are not always explicitly stated. Examples:

1. First line of Quantitative real-time PCR analysis - "Total RNA was extracted..."
2. First line of Western blotting- "Tissue homogenates were prepared..."

iii. Details of Western blot protocol/conditions (or references for more detailed protocol) not provided; full uncropped blots with ladder were also not provided

iv. Tracer experiment

1. Significance of cholesterol transport/metabolism should be addressed in Introduction- it isn't mentioned at all except for the final sentence of Introduction stating what your study found (when really there should be a focus on hypothesis/study intent in that section). The cholesterol tracer experiment therefore comes as a surprise in the Methods section.
2. Free cholesterol content is being measured in which tissue?

v. Sample homogenization- There are no fewer than 4 different experiments described by the authors that use tissue homogenate prepared in different conditions. The authors should explicitly state whether the homogenate samples processed were from different animals, or whether tissue was aliquoted from the same animals for the different homogenate preps. This could perhaps be included in the Sample Preparation subsection of Methods on Page 15.

vi. Statistics. I am not certain that mean +/- SEM is the most appropriate/accurate presentation of non-parametric statistical analyses of non-normally distributed data. Please verify with statistician.

c. Figures. Many areas to be considered for clarification and/or Figure improvement. In general there are so many figures and they are split between those included within the manuscript and those included as Supplemental. I found it very confusing to do so much back and forth. I would strongly encourage consolidating figures when able, and really consider carefully how to organize those which can be Supplemental as opposed to within the manuscript text, to avoid the reader feeling as though they are missing out on too much by not looking for Suppl Figures, and to avoid a ton of back and forth within a specific section.

Areas of Figure clarification-

i. Figure 1.

1. To me, it would be more intuitive to organize the various graphs of Figure 1 in order of development. Ie from blastocyst through to body weights at week 5-8. Might want to consider re-ordering.
2. I don't understand why a Supplemental Figure was needed to include the CRL for the B-kl+/+ wild types. Could those not be included in the existing graphs of Fig 1? Why are the sample sizes from the heterozygotes and knockouts in Fig 1C different from the sample sizes from these same groups in Fig S1B?
3. 1D is missing some explanation. The table below the images itself does not stand alone and the legend is not descriptive enough. Please define birth ratio/birth rate for us and provide labels for the numbers included (ie the 45 and 51- is this total pups?)

2- Errors throughout Figures & Figure Legends. In and of themselves, these individual errors are quite minor; however, with there being so many of them, to me it denotes a bigger picture oversight in the time taken to ensure accuracy of Figures and legends worthy of publication. See specific examples to revise:

- a. Figure 2C. The stars labeled on the graphs for Ldlr and Scarb1 (labeled as * for each) do not match the Figure legend describing them as having p values worthy of **.
- b. Figure 2D, 2E. The titles of these Figures and their description within the manuscript text match, but the Figure legends appear to have switched the "Embryo" and "Yolk Sac" designations. Additionally, if embryos and yolk sacs came from the same animals for these TLC animals, please ensure the sample sizes in the legend match between D and E. It looks like they may have been switched in legend for Fig 2E (n of 8 and 7, rather than 7 and 8 as described in legend for Fig 2D).
- c. Figure 3E. P value assigned on the graph for Lipc (**) does not match the p value description in the Figure legend (***).
- d. Figure 4E. The figure legend mentions that 3 representative images are included (including one for Fgf15+/-) but only Fgf15+/+ and -/- are shown in the figure.
- e. Figure 4F. P value assigned on the graph for Scarb1 (***) does not match the p value description in the Figure legend (**).

3- Organization of manuscript. In addition to the specific examples of clarification needed (including incorporation of study hypothesis/aim/goal) and confusion in finding the many figures, one of the bigger picture issues I have is in the authors using the Results section to incorporate their speculation and conclusions. The place for those types of commentary and ability to further expand on this commentary is in the Discussion, which would be enriched by including those statements in the Discussion and expanding on them/placing them in the context of FGR

MINOR POINTS

- Ex vivo LDL uptake in yolk sac experiment using BODIPY was not described in the general Methods section (and the separate write up included in the Supplemental Information was not referenced within the body of the manuscript)

- More detailed Figure legends in general, but specifically supplemental Fig S2
- Figure 3A and Figure S3A-B seem to say the same thing, except the supplemental version includes the wild type B-kl $+/+$.
 - o Why are separate figures needed?
 - o Unless I am misreading/interpreting, the sample sizes are different and the data values are different between the figures, even though I can't tell why these graphs should be displaying different information. Aren't they both displaying mRNA levels of B-kl in embryos and yolk sacs at E9.5?
- Of the Supplemental Tables, only Table S1 is actually labeled as such (the others have no labels)
- Please make sure that the Supplemental figures are also correctly referencing the appropriate regular figures (they seemed off to me in their legend titles).

June 18, 2023

Life Science Alliance manuscript #LSA-2023-01916-T

Dr. Novella Guidi,
Scientific Editor, *Life Science Alliance*

Dear Dr. Guidi

Please find attached a revised version of our manuscript "Feto-maternal cholesterol transport regulated by β -Klotho-FGF15 axis is essential for fetal growth", which we would like to resubmit for publication as a Research Article in *Life Science Alliance*. The comments by the reviewers were extremely helpful in enabling us to improve the quality of our manuscript. We particularly appreciate the very careful reading of our paper by Reviewer #2, who made many detailed suggestions.

In response to the reviewers' questions and suggestions, we have carefully revised the text, including adding the results of additional experiments. We believe that the new data, the revisions in the manuscript, and our explanations in the point-by-point responses given below address all the concerns raised by the reviewers. The major changes are as follows:

1. We modified the title to follow the character limit.
2. We compared the expression levels of genes involved in lipid metabolism in embryonic tissues and in adult tissues. The results further support our conclusion that β -KL is required to regulate lipid metabolism in embryos.
3. We newly evaluated the lipid contents in the placenta and found that the lipid levels were comparable in β -kl^{+/-} and β -kl^{-/-} placentas. This further supports our conclusion that β -KL expressed in yolk sac, not placenta, is essential for feto-maternal lipid transport.
4. In line with the suggestion by Reviewer #2, we have reorganized the panels in Figure 1 in order of development. This arrangement makes our data much easier to follow.
5. We have employed non-parametric statistical analysis of the data, with the exception of the body weight comparison. We have also used box-plot style to present the results. The reanalysis does not affect our conclusions.
6. Based on his contribution to this work, we have also credited Yo-ichi Nabeshima as another corresponding author.

We hope you will agree that the revised manuscript is now suitable for publication in *Life Science Alliance*, and we look forward to hearing from you at your earliest convenience.

Yours sincerely,

Kanako Kobayashi, Ph. D., Yo-ichi Nabeshima, M.D., Ph. D.

Responses to the comments by Reviewer #1

The authors here focus on a mouse model for fetal growth restriction (FGR). Fetal growth restriction is a complication of pregnancy with lasting effects for the offspring, clearly warranting research in this area. The authors use a genetic model where beta Klotho (β -KL) has been deleted. The β -kl-ko mice had already previously been shown to be growth restricted. β -KL is known to be expressed in the placenta at low levels, but in the yolk sac at high levels. The authors nicely demonstrate that FGR here occurs in the first half of pregnancy (when the yolk sac is more active than the placenta) and that cholesterol supply is also impaired.

This manuscript is clearly written and follows a logical strategy to characterize the phenotype. However, I have several concerns:

- In the human clinical situation, there are differences between boys and girls regarding the severity of FGR as well as the long-term outcome. Can the authors give some details about sex differences in their study? For example, was the ratio male/female the same in all genotypes, and how does the growth curve of the adolescent mice (Fig 1a) look like when split by sex?

Response-1:

In response to the reviewer's comment, we reanalyzed the genotype ratio and the crown-rump length (CRL) at E9.5 separately in females and males. We found no significant difference in the male/female ratio in both genotypes (Revised Table S1, $P=0.8151$ by chi-square test). Reduced CRL was observed in both female and male β -kl-/- embryos (Revised Fig. S1B and S1C). Based on our previous study (*FASEB J* 2016, 30, 849-862), we evaluated the growth curve only in male mice in the postnatal stages (Fig. 1A, Revised Fig. 1F). However, it was previously shown that male and female β -kl-/- mice both exhibit reduced body weight from 2 to 6 weeks of age to a similar extent (*JCI* 2005, 115, 2202-2208). These data suggest that the contribution of β -KL to growth is not sex-dependent, but further work will be needed to examine the long-term outcome. We have added a comment on this issue in the revised manuscript.

Added sentences in the revised manuscript:

(Results) Line 108 to 110

The body weight of β -kl-/- mice was lower in both female and male animals after birth (Ito et al., 2005). Likewise, no sex bias was observed in CRL at E9.5 (Fig. S1B and S1C).

(Discussion) Line 337 to 344

In the present study, no sex bias was observed in the severity of FGR in β -kl/- embryos. Further, reduced body weight in postnatal β -kl/- mice was seen similarly in females and males (Ito et al., 2005). These data suggest that the contribution of β -KL to growth is independent of gender. On the other hand, phenotypic assessments of β -kl/- mouse metabolism have mainly been done using male mice. In this context, the sex differences in the long-term consequences of FGR have been reported both in animal models and humans (Eriksson et al., 2010, Sohi et al., 2010, Cheong et al., 2016). Thus, we need to include female mice in future studies to determine the metabolic outcomes of FGR in β -kl/- mice.

- Fetal and yolk sac gene expressions are interesting, but what do they mean? Maybe the changes in expression in Fig 2-4 occur on a very low overall level? Can the authors show how high these expressions are compared to adult liver, for example? To judge the meaning of changes for example in bile acid metabolism (*Cyp7a1*) this would be helpful, as to the best of my knowledge bile acids are not produced that early in life.

Response-2:

Thank you for this important comment. To address this point, we newly compared the expression levels of genes involved in lipid regulation in fetal tissues and adult tissues. First, the mRNA levels of lipid metabolism-related genes in the embryos were comparable to or even higher than those in the adult liver (Revised Fig. S2B and S2C). These results support our conclusion that the transcriptional changes revealed by the pathway analyses were associated with the metabolic alterations in β -kl/- embryos. Second, expression of both *Cyp7a1* and *Cyp7b1* in the embryos and yolk sacs was almost undetectable, as the reviewer expected (Revised Fig. S2D). We have included these findings in the Results section. Third, lipase mRNAs were detectable in the E9.5 yolk sac, though at lower levels compared to those in the adult liver (Fig. S4). Although the physiological significance of the lipases in yolk sac remains unclear, increased expression of lipases might lead to some degree of lipid accumulation in β -kl/- yolk sacs.

Added sentences in the revised manuscript:

(Results) Line 131 to 149

To judge the significance of changes in transcript levels, we compared the expression levels of genes related to lipid metabolism in fetal tissues from control β -kl/+ embryos to those in adult livers (Fig. S2A). At E9.5, the level of 3-hydroxy-3-methylglutaryl-CoA reductase (*Hmgcr*) was higher in embryos than in adult liver, whereas 3-hydroxy-3-methylglutaryl-CoA synthase 1 (*Hmgcs1*) and the fatty acid synthase (*Fasn*) levels were comparable (Fig. S2B). Low-density lipoprotein (LDL) receptor (*Ldlr*) expression in embryos was comparable to that in adult liver (Fig. S2C). In contrast to

Ldlr, high-density lipoprotein (HDL) receptor (*Scarb1*) expression was markedly higher in yolk sacs than in embryos (Fig. S2C).

We also examined the mRNA levels of enzymes in synthesizing bile acids, since β -KL is indispensable for suppressing bile acid synthesis in adult mice (Ito et al., 2005, Tomiyama et al., 2010, Kobayashi et al., 2016). In the liver of adult β -*kl*^{-/-} mice, the expression of *Cyp7a1*, a rate-limiting enzyme of hepatic bile acid synthesis, is strongly upregulated, but *Cyp7a1* expression was undetectable in E9.5 embryos and yolk sacs by qPCR (data not shown). It is believed that the alternative pathway is the main contributor to production of bile acids in the fetal liver (Jia et al., 2021). However, to our surprise, the level of *Cyp7b1*, a gatekeeper to the alternative pathway, was much lower in E9.5 embryos than in the adult liver (Fig. S2D). Furthermore, *Cyp7b1* was expressed at similar levels in β -*kl*^{-/-} and β -*kl*^{+/-} embryos (Fig. S2E), contrary to our expectation that β -KL would be required to suppress the alternative pathway in the embryo.

- On page 6 the authors speculate about compensatory lipid synthesis, based on the TLC data. They should also consider the size of the yolk sac vs the embryo, as this might influence their conclusions. Also the placenta could be used as a control, because if the authors' hypothesis is correct it should show no differences.

Response-3:

We appreciate these helpful suggestions, and have added new data. First, to take account of the difference in size between yolk sac and embryo, we compared the tissue size of each TLC sample by using total protein amount as a surrogate indicator (Revised Fig.S3A). At E9.5, the total protein amount of the embryos was higher than the yolk sacs, though the difference was not significant in the control heterozygous (Revised Fig. S3A). We calculated the lipid levels per tissue by multiplying protein amount per tissue (whole embryo or yolk sac) by the lipid levels per μ g of protein. On this basis, the contents of cholesteryl esters (CE) were markedly higher in the yolk sac than in the embryo, though the levels of triglycerides (TG) were comparable (Revised Fig. S3B). These data strongly support a significant role of the yolk sac in lipid metabolism in the embryo. Second, we newly evaluated the lipid contents in the placenta by TLC. We found no significant difference in CE or TG contents between β -*kl*^{+/-} and β -*kl*^{-/-} placentas (Revised Fig. S3C). These results also support our conclusion that β -KL expressed in the yolk sac, not the placenta, is important for lipid regulation in the embryo.

Added sentences in the revised manuscript:

(Results) Line 155 to 167

The murine yolk sac is a membranous sac surrounding the embryo (Zohn and Sarker, 2010), and its size as a tissue size is smaller than that of the whole embryo. Indeed, we confirmed this in the TLC

samples by using total protein amount as a surrogate indicator (Fig. S3A). Surprisingly, however, the CE content per tissue was higher in the yolk sacs than in the embryos, while TG levels were comparable in yolk sacs and embryos (Fig. S3B). β -kl^{-/-} yolk sacs were significantly smaller than β -kl^{+/-} yolk sacs (Fig. S3A), but no difference was observed in the lipid levels per yolk sacs (Fig. S3B). Whether assessed per μ g of protein or per tissue, β -kl^{-/-} embryos showed significantly reduced lipid levels (Fig. 2D and Fig. S3B). We also measured the lipid contents in the placenta, because β -kl mRNA was detected in the E12.5 placenta, even though at markedly lower levels than in the yolk sac or fetal liver (Fig. S3C). We found that the lipid contents were similar in β -kl^{-/-} and β -kl^{+/-} placentas (Fig. S3D). These results indicate that β -KL is required to regulate lipid metabolism in the embryo in the yolk sac-dependent period.

- It is indeed fascinating that this very early event has such long-lasting consequences. However, the discussion goes too far by saying "... suggesting that cholesterol shortage during the embryonic period has a lifelong impact on the body size." as the authors cannot exclude that other molecules mediate this effect (for example oxysterols or bile acids).

Response-4:

We agree, and have deleted the statement. As you pointed out, we were not able to exclude the involvement of other molecules in the current study. In regard to bile acids, our previous study had shown that reduced body weight in β -kl^{-/-} mice was not the consequence of the elevated bile acid levels (*FASEB J* 2016, 30, 849-862).

Minor concerns:

- In the introduction, the statements about long term consequences of FGR (Barker hypothesis) and FGR itself are a bit mixed, this could be better structured. I would also prefer to have clearly defined aims rather than indirectly mentioned ones.

Response-5:

We have rewritten the Introduction section to better explain the aims of this work.

Added sentences in the revised manuscript:

(Introduction) Line 86 to 88

We hypothesized that β -KL expressed in yolk sac is essential for metabolic regulation in the embryo, and consequently would have an impact on fetal growth and body size in postnatal life. To test this hypothesis, we analyzed β -kl^{-/-} embryos at the transcript and metabolite levels.

Deleted sentences in the revised manuscript:

(Introduction) p.3 Line 2 to 6

Metabolic syndrome refers to the coexistence of multiple cardiovascular risk factors, such as type 2 diabetes, dyslipidemia, and hypertension (Eckel et al., 2010). Although excess eating and a sedentary lifestyle are the leading causes of these obesity-associated diseases, epidemiological studies have revealed that low birth weight is inversely related to midlife dysmetabolism (Barker 2007, Hsu et al., 2021).

- A vaginal plug was used as a sign of successful mating - does this mean that the males were with the females for more than one night? How can you make sure that timing is precise?

Response-6:

Males and females were cohoused for only one night, and there was an interval of several days before female mice were used for the next mating. In addition, we counted the number of somites and confirmed that there was no difference in the somite number among the genotypes (Revised Fig. 1D).

Added sentences in the revised manuscript:

(Materials and Methods) Line 372 to 373

To obtain embryos, one male mouse was allowed to cohabit with one or two female mice overnight.

Responses to the comments by Reviewer #2

This manuscript by K. Kobayashi et al. attempts to demonstrate that the B-KL/FGF axis regulates feto-maternal cholesterol transport through the yolk sac, and that its dysfunction leads to FGR in mice. The study uses a B-KL knockout and a range of methods to study the underlying mechanisms explaining the link between altered B-KL function and fetal growth. Overall this study could demonstrate important findings to better understand FGR mechanistically. However, in its current form the study lacks an explicitly stated central hypothesis, the Methods are incomplete (and experiments therefore not reproducible), the Results read more like a combination of Methods and Discussion, and the Discussion itself is underdeveloped. There are tons of figures and a robust Supplemental Figure library that at times is confusing to toggle to from the general manuscript and a host of simple errors in the consistency between Figures/Figure legend. All of these nuances made it difficult to focus on the underlying science in order to completely comment on the impact the study results might have on the field. Major revisions are needed in order for another review to ascertain readiness for publication.

Response-7:

We appreciate the reviewer's detailed and constructive comments on our manuscript. We have extensively modified the manuscript in response, as described in detail below.

MAJOR POINTS

1- Lack of clarity and/or reproducible detail

a. Hypothesis/study goal. Neither of these are explicitly stated in either the Abstract or Introduction. The closest the authors come to stating a hypothesis is found on page 8 in Results subsection Deletion of B-KL leads to lipid abnormalities in the yolk sac, with the sentence starting with "Since the embryo absorbs cholesterol..." A hypothesis really should be explicitly stated earlier in the paper to provide the framework for all other sections, and should be re-visited in the Discussion section.

Response-8:

β -KL regulates hepatic bile acid synthesis in adult animals, but its role during development is not known. The aim of our study was to determine the role of β -KL in the developmental stages. In the revised manuscript, we have stated this in the Abstract and Introduction, and refocused the Discussion.

Added sentences in the revised manuscript:

(Abstract) Line 44 to 45

We hypothesized that β -KL is required for metabolic regulation in the embryo and aimed to clarify the role of β -KL during development.

(Introduction) Line 86 to 88

We hypothesized that β -KL expressed in yolk sac is essential for metabolic regulation in the embryo, and consequently would have an impact on fetal growth and body size in postnatal life. To test this hypothesis, we analyzed β -kl^{-/-} embryos at the transcript and metabolite levels.

(Discussion) Line 284 to 288

Thus, the aim of this study was to determine the role of β -KL in the developmental stages. Specifically, we hypothesized that β -KL expressed in yolk sac, the predominant site of β -kl expression in fetal tissues, is required for metabolic regulation in the embryo, consequently controlling fetal growth and postnatal body size.

b. Methods. There is much detail lacking in this section that would help reviewers and readers. Experiments should be reproducible, but there is not enough detail or references provided for this to be possible. See specific examples below:

i. Animals-

1. No mention of the type of mouse used, until it is mentioned in Results supplementary Fig S1A legend (C57BL6/J).

Response-9:

We apologize for the insufficient descriptions. We have added further details throughout the Methods section, including information on the C57BL6/J mice used in this study.

Added sentences in the revised manuscript:

(Materials and Methods) Line 371 to 372

C57BL/6JJcl mice were purchased from CLEA Japan (Tokyo, Japan).

2. No mention of the embryonic day time points at which animals were sacrificed/tissues collected for the various experiments. We learn in the results section what these time points are, but should be included in Methods, along with some mention of the significance of selected time points (ie. E9.5 and E12.5).

Response-10:

We have added the requested details.

Added sentences in the revised manuscript:

(Materials and Methods)

Sample preparation (Line 380 to 383)

Based on the milestones of intrauterine development in mice (Cross et al., 1994, Rossant and Cross 2001), tissues were collected at E3.5, E9.5 and E12.5, corresponding to preimplantation, yolk sac-dependent, and placenta-dependent stages, respectively.

RNA-sequencing analysis (Line 396 to 398)

Total RNA was extracted from E9.5 embryos using a RNeasy Mini kit (QIAGEN) and treated with DNase I (QIAGEN) on-column according to the manufacturer's instructions.

Quantitative real-time PCR analysis (Line 412 to 415)

Total RNA was extracted from whole embryo (E9.5), yolk sac (E9.5 or E12.5), liver (E12.5), or placenta (E12.5) using a RNeasy Mini Kit (QIAGEN) or AllPrep DNA/RNA/Protein Mini Kit (QIAGEN).

Western blotting (Line 424 to 428)

Tissue homogenates of yolk sac (E9.5) and adult liver (male β -*kl*^{+/-} mice, 12 weeks of age) were prepared in buffer [20 mM 4-(2-hydroxyethyl)-1-piperazineethanesulfonic acid, 150 mM NaCl, and 0.5% Nonidet P-40 (pH 7.4)] containing protease inhibitor cocktail (Roche, cOmplete, Mini, EDTA-free), and phosphatase inhibitor cocktail (Nacalai tesque).

Thin-layer chromatography (Line 439 to 441)

Whole embryo and yolk sac without allantois were collected at E9.5. Allantoises were used for genotyping. Placentas were dissected at E12.5.

Tracer experiment (Line 457 to 461)

One milligram of [2,3,4-¹³C]-cholesterol (Cambridge Isotope Laboratories) dissolved in 40 μ l of ethanol was mixed with 10% Intralipos (Otsuka Pharmaceutical) (1:1, v/v) just before administration, and the lipid solution was intraperitoneally injected into pregnant β -*kl*^{+/-} mice at E8.5. Then, the whole embryos were collected at E9.5 under microscopic observation.

Histologic analysis (Line 471 to 472)

Yolk sacs (E9.5) were fixed in 10% formalin (Wako) and embedded in paraffin.

LDL uptake assay (Line 482 to 484)

E9.5 embryo with yolk sac was dissected in PBS and incubated in 75% lipoprotein-deficient fetal calf serum (LPDS) (Sigma) and 25% DMEM for 30 min at 37°C in 5% CO₂.

ii. Specific tissue types for each experiment are not always explicitly stated. Examples:

1. First line of Quantitative real-time PCR analysis - "Total RNA was extracted..."

Response-11:

We have added details of the types of tissues used for each experiment.

2. First line of Western blotting- "Tissue homogenates were prepared..."

Response-12:

We have added further details.

iii. Details of Western blot protocol/conditions (or references for more detailed protocol) not provided; full uncropped blots with ladder were also not provided

Response-13:

We have added details of the conditions in the Methods section. The uncropped blots are shown below. The same images have been submitted as Source Data.

Added sentences in the revised manuscript:

(Materials and Methods) Line 428 to 437

The supernatant was electrophoresed on 4-20% gradient SDS-polyacrylamide gel and transferred to PVDF membranes. After blocking with 5% skim milk in TBS-T for 1 h at room temperature, the membrane was incubated overnight with goat anti-mouse KLB polyclonal antibody (R&D Systems AF2619, 1:1,000) at 4°C. Then the membrane was incubated with anti-goat HRP (Jackson 705-035-147, 1:100,000) for 45 min at room temperature. The signals were developed by using the ECL Prime Western Blotting Detection Reagents (Cytiva) and detected with a FUSION SYSTEM (Vilber-Lourmat). After stripping, the membrane was reprobred with rabbit anti-β-Actin monoclonal antibody (Cell Signaling 4970, 1:1,000) and anti-rabbit HRP (Thermo A16110, 1:200,000).

iv. Tracer experiment

1. Significance of cholesterol transport/metabolism should be addressed in Introduction- it isn't mentioned at all except for the final sentence of Introduction stating what your study found (when really there should be a focus on hypothesis/study intent in that section). The cholesterol tracer experiment therefore comes as a surprise in the Methods section.

Response-14:

We agree, and have added the following sentences in the Introduction section.

Added sentences in the revised manuscript:

(Introduction) Line 61 to 64

For example, tracer experiments using stable isotope-labeled cholesterol showed that cholesterol derived from maternal blood is transported to embryos through the yolk sac in mice (Yoshida and Wada, 2005).

2. Free cholesterol content is being measured in which tissue?

Response-15:

We measured free cholesterol in the whole embryos. We have added the sample information in the Methods section.

Added sentences in the revised manuscript:

(Materials and Methods) Line 464 to 466

Free cholesterol contents in each whole embryo with/without [2,3,4-¹³C] labeling were determined by GC-MS (GCMS-QP2010Ultra, Shimadzu) as previously described (Yoshida and Wada, 2005).

v. Sample homogenization- There are no fewer than 4 different experiments described by the authors that use tissue homogenate prepared in different conditions. The authors should explicitly state whether the homogenate samples processed were from different animals, or whether tissue was aliquoted from the same animals for the different homogenate preps. This could perhaps be included in the Sample Preparation subsection of Methods on Page 15.

Response-16:

Thank you for raising this point. Due to the small size, it was impractical to split the homogenate from one animal for multiple experiments. Therefore, we prepared tissue homogenates for each experiment from different animals. We have added an explanation about the sample preparation.

Added sentences in the revised manuscript:

(Materials and Methods) Line 393 to 394

Each sample was prepared from a different animal, unless mentioned in the figure legends.

vi. Statistics. I am not certain that mean +/- SEM is the most appropriate/accurate presentation of non-parametric statistical analyses of non-normally distributed data. Please verify with statistician.

Response-17:

As you suggested, we consulted a statistician, and as a result, we have applied non-parametric analyses to all data obtained from fetal samples. In addition, the graphs have been changed to boxplot format. In line with our previous studies, we employed parametric statistical analysis of the

body weight changes in the postnatal period (Revised Fig. 1F). Details of the statistical analysis have been included in the figure legends, according to the journal's guidelines. P-values are shown in Source Data File for each figure.

Added sentences in the revised manuscript:

(Materials and Methods) Line 497 to 502

Statistics

All statistical analyses were performed with EZR (version 1.52) (Kanda 2013). Statistical analyses were performed using a 2-tailed t-test or Mann-Whitney's U-test (2 groups) as indicated in figure legends. Bonferroni correction was applied for multiple-group analysis (3 or more groups). $P < 0.05$ was considered significant. The number of replicates is given in each figure legend.

c. Figures. Many areas to be considered for clarification and/or Figure improvement. In general there are so many figures and they are split between those included within the manuscript and those included as Supplemental. I found it very confusing to do so much back and forth. I would strongly encourage consolidating figures when able, and really consider carefully how to organize those which can be Supplemental as opposed to within the manuscript text, to avoid the reader feeling as though they are missing out on too much by not looking for Suppl Figures, and to avoid a ton of back and forth within a specific section. Areas of Figure clarification-

i. Figure 1.

1. To me, it would be more intuitive to organize the various graphs of Figure 1 in order of development. I.e from blastocyst through to body weights at week 5-8. Might want to consider re-ordering.

Response-18:

Thank you for this very helpful suggestion. We have reorganized the figures in line with your advice. In Figure 1, we have placed the graphs in order of development (Revised Fig. 1A-1F). As we employed β -kl+/+ embryos as a control in Revised Fig. 1C and 1D, we relocated the photos (Fig. 1D) to Supplemental Fig. S1D. We revised the Results section accordingly.

Added sentences in the revised manuscript:

(Results) Line 96 to 119

To evaluate the impact of β -KL on growth, we examined the size of β -kl/- mice through the developmental stages to the postnatal period. We first compared the size and cell numbers of blastocysts at E3.5 among the genotypes. Before implantation, no difference was observed in the size or number of cells in β -kl/- blastocysts compared with β -kl+/+ and β -kl+/- blastocysts (Fig. 1A and 1B).

Murine embryos absorb maternal nutrients through the yolk sac until E10 (Zohn and Sarkar 2010). Hence, we next examined the size of embryos at E9.5 as a representative point in the yolk sac-dependent period. The crown-rump length (CRL) of β -kl^{-/-} embryos was significantly shorter than that of β -kl^{+/-} embryos at E9.5 (Fig. 1C). The somite numbers were comparable among the three genotypes (Fig. 1D). Since β -kl^{+/+} and β -kl^{+/-} embryos were indistinguishable in terms of growth at E3.5 and E9.5, we used β -kl^{+/-} embryos as a control to provide a sufficient number of embryos for experiments. We again confirmed that β -kl^{-/-} embryos were significantly smaller than β -kl^{+/-} embryos in different cohorts (Fig. S1A). The body weight of β -kl^{-/-} mice was lower in both female and male animals after birth (Ito et al., 2005). Likewise, no sex bias was observed in CRL at E9.5 (Fig. S1B and S1C). On the other hand, the birth ratio of β -kl^{-/-} mice was less than expected from Mendelian inheritance (Ito et al., 2005, Somm et al., 2017), though the embryonic genotypes were segregated to approximately Mendelian ratios at E9.5 (Table S1). The gross appearance of β -kl^{-/-} embryos was normal at E9.5 (Fig. S1D). We also evaluated the CRL at E12.5 and confirmed that β -kl^{-/-} embryos remained smaller than β -kl^{+/-} embryos during the placenta-dependent period (Fig. 1E).

Consistent with our previous report (Kobayashi et al., 2016), the body weight of β -kl^{-/-} mice remained significantly lower than that of their control littermates (Fig. 1F). These findings demonstrate that β -KL is required for growth in the post-implantation embryos, but is not involved in morphogenesis or survival up to E9.5.

2. I don't understand why a Supplemental Figure was needed to include the CRL for the B-kl^{+/+} wild types. Could those not be included in the existing graphs of Fig 1? Why are the sample sizes from the heterozygotes and knockouts in Fig 1C different from the sample sizes from these same groups in Fig S1B?

Response-19:

We apologize for the insufficient explanation. The data shown in Fig. 1C and Fig. S1B were obtained from independent cohorts, and the sample sizes were indeed different. The reason why we included Fig. 1C in the regular figure instead of Fig. S1B was that we mainly used β -kl^{+/-} heterozygotes as a control in this study. Now, we have relocated Fig. S1B to the regular figure as Fig. 1C (Revised Fig. 1C). Along with this, we have added an explanation of why we used heterozygotes as a control in the Results sections. We have also indicated the genotype of mice used to obtain embryos in each figure legend.

Added sentences in the revised manuscript:

(Results) Line 105 to 107

Since β -kl+/+ and β -kl+/- embryos were indistinguishable in terms of growth at E3.5 and E9.5, we used β -kl+/- embryos as a control to provide a sufficient number of embryos for experiments.

3. 1D is missing some explanation. The table below the images itself does not stand alone and the legend is not descriptive enough. Please define birth ratio/birth rate for us and provide labels for the numbers included (ie the 45 and 51- is this total pups?)

Response-20:

In the revised manuscript, we have repositioned the table of birth ratio as Supplemental Table S1 and added further details - 45 and 51 were the numbers of embryos of each genotype.

Added sentences in the revised manuscript:

(Revised Table S1)

Table S1. Summary of β -kl genotypic segregation at E9.5

	β -kl+/-		β -kl-/-		Total
	♂	♀	♂	♀	
Number of embryos	25	20	29	22	96
%	26.0	20.8	30.2	22.9	100.0

Embryos were obtained from β -kl-/- female mice (n = 12) crossed with β -kl +/- male mice.

2- Errors throughout Figures & Figure Legends. In and of themselves, these individual errors are quite minor; however, with there being so many of them, to me it denotes a bigger picture oversight in the time taken to ensure accuracy of Figures and legends worthy of publication. See specific examples to revise:

a. Figure 2C. The stars labeled on the graphs for Ldlr and Scarb1 (labeled as * for each) do not match the Figure legend describing them as having p values worthy of **.

Response-21:

We apologize for the errors in figures and legends. We have carefully checked all of them and corrected various mistakes, including those you kindly pointed out.

b. Figure 2D, 2E. The titles of these Figures and their description within the manuscript text match, but the Figure legends appear to have switched the "Embryo" and "Yolk Sac" designations.

Additionally, if embryos and yolk sacs came from the same animals for these TLC animals, please ensure the sample sizes in the legend match between D and E. It looks like they may have been switched in legend for Fig 2E (n of 8 and 7, rather than 7 and 8 as described in legend for Fig 2D).

Response-22:

Thank you. We have corrected it. As the reviewer thought, the embryos and yolk sacs used in the TLC analyses were obtained from the same animals. Although we prepared TLC samples at n = 8 for each genotype, one β -kl^{-/-} embryo and one β -kl^{+/-} yolk sac sample were not properly applied to the TLC plate, and we excluded these two samples from the data. That is why the sample sizes were different between Fig. 2D and Fig. 2E.

c. Figure 3E. P value assigned on the graph for Lipc (***) does not match the p value description in the Figure legend (***).

Response-23:

We have corrected it.

d. Figure 4E. The figure legend mentions that 3 representative images are included (including one for Fgf15^{+/-}) but only Fgf15^{+/+} and ^{-/-} are shown in the figure.

Response-24:

Thank you. We have corrected it.

e. Figure 4F. P value assigned on the graph for Scarb1 (***) does not match the p value description in the Figure legend (**).

Response-25:

We have corrected it.

3- Organization of manuscript. In addition to the specific examples of clarification needed (including incorporation of study hypothesis/aim/goal) and confusion in finding the many figures, one of the bigger picture issues I have is in the authors using the Results section to incorporate their speculation and conclusions. The place for those types of commentary and ability to further expand on this commentary is in the Discussion, which would be enriched by including those statements in the Discussion and expanding on them/placing them in the context of FGR

Response-26:

We agree, and have reorganized the manuscript, moving the speculations/conclusions from the Results section to the Discussion.

MINOR POINTS

- Ex vivo LDL uptake in yolk sac experiment using BODIPY was not described in the general Methods section (and the separate write up included in the Supplemental Information was not referenced within the body of the manuscript)

Response-27:

We apologize for the inappropriate placement of the experimental procedure. We have relocated “LDL uptake assay” to the general Methods section (Line 482 to 495).

- More detailed Figure legends in general, but specifically supplemental Fig S2

Response-28:

As suggested, we have added further details in the revised Figure legends, including Fig. S2 (now Fig. S5 in the revised manuscript).

Added sentences in the revised manuscript:

(Revised Fig. S5) Line 797 to 815

Figure S5, related to Figure 3. Yolk sac β -KL is required for lipid regulation in E9.5 embryo
(A) β -kl mRNA levels in E9.5 embryos (β -kl^{+/-}, n = 9; β -kl^{-/-}, n = 8; β -kl^{+/-}/Tg, n = 8; β -kl^{-/-}/Tg, n = 9). **(B)** β -kl mRNA levels in E9.5 yolk sacs (β -kl^{+/-}, n = 9; β -kl^{-/-}, n = 8; β -kl^{+/-}/Tg, n = 8; β -kl^{-/-}/Tg, n = 9). **(C)** Crown-rump length in embryos at E9.5 (β -kl^{+/-}, n = 9; β -kl^{-/-}, n = 7; β -kl^{+/-}/Tg, n = 7; β -kl^{-/-}/Tg, n = 9). **(D)** *Hmgcr* and *Ldlr* mRNA levels in E9.5 embryos (β -kl^{+/-}, n = 10; β -kl^{-/-}, n = 8; β -kl^{+/-}/Tg, n = 8; β -kl^{-/-}/Tg, n = 9). In **(A)** and **(D)**, data are shown as fold increase over the average expression levels in β -kl^{+/-} embryos. **(E)** *Lpl* RNA levels in E9.5 yolk sacs (β -kl^{+/-}, n = 9; β -kl^{-/-}, n = 8; β -kl^{+/-}/Tg, n = 8; β -kl^{-/-}/Tg, n = 9). In **(B)** and **(E)**, data are shown as a fold increase over the average expression levels in β -kl^{+/-} yolk sacs. **(F)** Experimental design for ex vivo LDL uptake assay at E9.5. **(G)** Wholemount images of the yolk sacs fixed following LDL uptake at 37°C for 30 min. Scale bars: 1 mm. **(H)** Confocal images of the yolk sacs. The sections were prepared from the yolk sacs using ex vivo LDL uptake experiment. BODIPY-FL signals were shown in green, and nuclei stained with DAPI were shown in cyan. Scale bars: 5 μ m. All samples were obtained from female β -kl^{+/-} mice mated with male β -kl^{-/-}/Tg mice.

Data information: In **(A-E)**, midlines represent the median, boxes the interquartile range (25th to 75th percentile), and whiskers the range of data. **P*<0.05, ***P*<0.01, ****P*<0.001 (Mann-Whitney's U-test with the Bonferroni correction).

- Figure 3A and Figure S3A-B seem to say the same thing, except the supplemental version includes the wild type β -kl +/+.
- o Why are separate figures needed?

Response-29:

We apologize for the confusion. In Figure 3A, we intended to show that β -kl mRNA levels in the yolk sacs were much higher than in embryos. In the revised manuscript we have replaced Figure 3A with newly obtained data to compare β -kl mRNA levels in the embryos and yolk sacs to those in adult liver (Revised Fig. 3A). On the other hand, Figures S3A and S3B, in combination with Figure S3C, were intended to show that one allele of β -kl is enough to regulate *Fgf15* mRNA levels in embryos. We have added an explanation in the revised manuscript and renewed the figures to clarify this.

Added sentences in the revised manuscript:

(Results) Line 170 to 173

To better understand the role of β -KL in lipid metabolism during development, we focused on β -KL expression in the yolk sac. At E9.5, the level of β -kl mRNA in the yolk sac was approximately 120-fold higher than that in the whole embryo (Fig. 3A). Notably, β -kl expression in the yolk sac was also 1.8-fold higher than that in adult liver.

(Results) Line 261 to 263

In this study, we mainly used β -kl^{+/-} embryos as a control. To examine whether one allele of β -kl was enough to suppress *Fgf15* expression, we measured the *Fgf15* levels in β -kl^{+/+} embryos and yolk sacs.

- o Unless I am misreading/interpreting, the sample sizes are different and the data values are different between the figures, even though I can't tell why these graphs should be displaying different information. Aren't they both displaying mRNA levels of β -kl in embryos and yolk sacs at E9.5?

Response-30:

We apologize for the insufficient explanation. Figure 3A and S3A-S3B both show β -kl mRNA levels in fetal tissues, but we obtained the data from independent cohorts. Hence, the sample sizes were different. We have added information about the genotypes of mice used to obtain samples in all figure legends.

- Of the Supplemental Tables, only Table S1 is actually labeled as such (the others have no labels)

Response-31:

Thank you for pointing this out. We have added titles to all tables.

Table S1 Summary of β -kl genotypic segregation at E9.5

Table S2 Results of k-means clustering

Table S3 Results of pathway analysis by the GAGE method

Table S4 List of TaqMan probes

- Please make sure that the Supplemental figures are also correctly referencing the appropriate regular figures (they seemed off to me in their legend titles).

Response-32:

We have checked and reorganized all associations between the regular figures and supplemental figures.

Figure S1, related to Figure 1. Growth of β -kl^{+/-} and β -kl^{-/-} embryos

Figure S2, related to Figure 2. Validation of the expression levels of the genes involved in lipid metabolism in fetal tissues

Figure S3, related to Figure 2. β -KL is required for lipid regulation in embryo and yolk sac

Figure S4, related to Figure 3. Validation of the mRNA levels of lipid-metabolizing enzymes in fetal tissues

Figure S5, related to Figure 3. Yolk sac β -KL is required for lipid regulation in E9.5 embryo

Figure S6, related to Figure 4. Expression profiles of genes involved in FGF signaling in β -kl and *Fgf15* mutants

July 16, 2023

RE: Life Science Alliance Manuscript #LSA-2023-01916-TR

Dr. Kanako Kobayashi
Kyoto University
Department of Aging Science and Medicine
53 Shogoin Kawahara-cho, Sakyo-ku
Kyoto 606-8507
Japan

Dear Dr. Kobayashi,

Thank you for submitting your revised manuscript entitled "Feto-maternal cholesterol transport regulated by β -Klotho-FGF15 axis is essential for fetal growth". We would be happy to publish your paper in Life Science Alliance pending final revisions necessary to meet our formatting guidelines.

- please address only the remaining Reviewer 2's point regarding the similarity of table S3 and figure 2B
- please add ORCID ID for the corresponding secondary author--they should have received instructions on how to do so
- please add the Twitter handle of your host institute/organization as well as your own or/and one of the authors in our system

Figure checks:

- please indicate the scale bar size in Legend for Figure S5 G and H

A. FINAL FILES:

B. MANUSCRIPT ORGANIZATION AND FORMATTING:

Sincerely,

Reviewer #1 (Comments to the Authors (Required)):

Thanks for addressing all my comments in a convincing way. Congratulations to this manuscript.

Reviewer #2 (Comments to the Authors (Required)):

In this revised version of K. Kobayashi et al's manuscript, newly entitled "Feto-maternal cholesterol transport regulated by B-Klotho-FGF15 axis is essential for fetal growth", the authors test the hypothesis that B-KL expression in the yolk sac in developing mice is essential for metabolic regulation in the embryo, impacting fetal growth. The authors have made SIGNIFICANT revisions to the original document, addressing every one of my comments which I believe have made this a stronger manuscript worthy of publication. Specifically, manuscript strengths now include: the authors' attention to appropriate statistical analysis, explicit statement of a hypothesis, reorganization of figures, and statements of transparency and clarity throughout the manuscript that aid in reader understanding behind the methods. It is now easier to see that the data are supportive of the authors' main points in the paper, and I have no further major revision suggestions.

MINOR POINTS

-There is still a large amount of data (central to the main points of the study) presented in supplemental figures, which I worry will not be sought out by the average reader for viewing. Perhaps there is a figure limitation by the journal that I am unaware of. But if able to incorporate more of the data into the main manuscript, I would make the effort!

-Though Table S3 provides the list of genes as additional information, the table overall seems to duplicate Figure 2B. Perhaps consider how best to display these data to avoid redundancy.

July 21, 2023

Life Science Alliance manuscript #LSA-2023-01916-TRR

Dr. Novella Guidi,
Scientific Editor, *Life Science Alliance*

Dear Dr. Guidi

Thank you for giving us the opportunity to publish our work in *Life Science Alliance*. Please find attached a final version of our manuscript "Feto-maternal cholesterol transport regulated by β -Klotho-FGF15 axis is essential for fetal growth". We responded to the points you mentioned as follows:

-please address only the remaining Reviewer 2's point regarding the similarity of table S3 and figure 2B

We thank the comments by Reviewer#2. To avoid redundancy, we have visualized the data in Fig.2B using bar-plot style, which is widely used to show the results of enrichment analysis. The change does not affect our conclusions.

-please add ORCID ID for the corresponding secondary author--they should have received instructions on how to do so

The corresponding secondary author added his ORCID ID.

-please add the Twitter handle of your host institute/organization as well as your own or/and one of the authors in our system

We have added the Twitter handle of our institute, Kyoto University. The authors are not on Twitter.

Figure checks:

-please indicate the scale bar size in Legend for Figure S5 G and H

We have added the scale bar size in the legend for Figures S5G (Line 804) and S5H (Line 807).

We hope you will agree that the final manuscript is suitable for publication in *Life Science Alliance*, and we look forward to hearing from you at your earliest convenience.

Yours sincerely,

Kanako Kobayashi, Ph. D., Yo-ichi Nabeshima, M.D., Ph. D.

July 26, 2023

RE: Life Science Alliance Manuscript #LSA-2023-01916-TRR

Dr. Kanako Kobayashi
Kyoto University
Department of Aging Science and Medicine
53 Shogoin Kawahara-cho, Sakyo-ku
Kyoto 606-8507
Japan

Dear Dr. Kobayashi,

Thank you for submitting your Research Article entitled "Feto-maternal cholesterol transport regulated by β -Klotho-FGF15 axis is essential for fetal growth". It is a pleasure to let you know that your manuscript is now accepted for publication in Life Science Alliance. Congratulations on this interesting work.

DISTRIBUTION OF MATERIALS:

Again, congratulations on a very nice paper. I hope you found the review process to be constructive and are pleased with how the manuscript was handled editorially. We look forward to future exciting submissions from your lab.

Sincerely,
